# Decapping protein EDC4 regulates DNA repair and phenocopies BRCA1

Gonzalo Hernández[1,2], María José Ramírez[1,2], Jordi Minguillón[1,2], Paco Quiles[3], Gorka Ruiz de Garibay[4], Miriam Aza-Carmona[1,2], Massimo Bogliolo[1,2], Roser Pujol[1,2], Rosario Prados-Carvajal[5], Juana Fernández[3], Nadia García[4], Adrià López[6], Sara Gutiérrez-Enríquez[7], Orland Diez[7,8], Javier Benítez[2,9], Mónica Salinas[3], Alex Teulé[3], Joan Brunet [ID] [3,6], Paolo Radice[10], Paolo Peterlongo[11], Detlev Schindler[12], Pablo Huertas [ID] [5], Xose S Puente[13], Conxi Lázaro[3], Miquel Àngel Pujana[4,14] & Jordi Surrallés[1,2,15]

BRCA1 is a tumor suppressor that regulates DNA repair by homologous recombination. Germline mutations in *BRCA1* are associated with increased risk of breast and ovarian cancer and BRCA1 deficient tumors are exquisitely sensitive to poly (ADP-ribose) polymerase (PARP) inhibitors. Therefore, uncovering additional components of this DNA repair pathway is of extreme importance for further understanding cancer development and therapeutic vulnerabilities. Here, we identify EDC4, a known component of processing-bodies and regulator of mRNA decapping, as a member of the BRCA1-BRIP1-TOPBP1 complex. EDC4 plays a key role in homologous recombination by stimulating end resection at double-strand breaks. EDC4 deficiency leads to genome instability and hypersensitivity to DNA interstrand cross-linking drugs and PARP inhibitors. Lack-of-function mutations in *EDC4* were detected in *BRCA1/2*-mutation-negative breast cancer cases, suggesting a role in breast cancer susceptibility. Collectively, this study recognizes EDC4 with a dual role in decapping and DNA repair whose inactivation phenocopies BRCA1 deficiency.

[1] Department of Genetics and Microbiology, Universitat Autònoma de Barcelona, Bellaterra 08193, Spain. [2] Centro de Investigación Biomédica en Red de Enfermedades Raras (CIBERER), Barcelona 08193, Spain. [3] Hereditary Cancer Programme, Catalan Institute of Oncology (ICO), Bellvitge Institute for Biomedical Research (IDIBELL), L'Hospitalet del Llobregat, Barcelona 08908, Spain. [4] Breast Cancer and Systems Biology Laboratory, Program Against Cancer Therapeutic Resistance (ProCURE), ICO, IDIBELL, L'Hospitalet del Llobregat, Barcelona 08908, Spain. [5] Centro Andaluz de Biología Molecular y Medicina Regenerativa (CABIMER) and Departamento de Genética, Universidad de Sevilla, Sevilla 41080, Spain. [6] Hereditary Cancer Programme, ICO, Girona Biomedical Research Institute (IDIBGI), Girona 17007, Spain. [7] Oncogenetics Group, Vall d´Hebron Institute of Oncology (VHIO), Barcelona 08035, Spain. [8] Area of Clinical and Molecular Genetics, Hospital Universitari Vall d'Hebron, Barcelona 08035, Spain. [9] Human Cancer Genetics Program, Spanish National Cancer Research Centre (CNIO), Madrid 28029, Spain. [10] Department of Preventive and Predictive Medicine, Unit of Molecular Bases of Genetic Risk and Genetic Testing, Fondazione IRCCS (Istituto Di Ricovero e Cura a Carattere Scientifico) Istituto Nazionale dei Tumori (INT), Milan 20133, Italy. [11] Department of Preventive and Predictive Medicine, Fondazione IRCCS Istituto Nazionale dei Tumori, IFOM, Fondazione Istituto FIRC di Oncologia Molecolare and Unit of Molecular Bases of Genetic Risk and Genetic Testing, Milan 20139, Italy. [12] Department of Human Genetics, Wurzburg University, Wurzburg 97070, Germany. [13] Department of Biochemistry and Molecular Biology, Instituto Universitario de Oncología, Universidad de Oviedo, Oviedo 33006, Spain. [14] Centro de Investigación Biomédica en Red de Oncología (CIBERONC), Oviedo 33006, Spain. [15] Department of Genetics and Biomedical Research Institute Sant Pau (IIB Sant Pau), Hospital de la Santa Creu i Sant Pau, Barcelona 08028, Spain. These authors contributed equally: Gonzalo Hernández, María José Ramírez. Correspondence and requests for materials should be addressed to M.A.P. (email: mapujana@iconcologia.net) or to J.S. (email: jsurralles@santpau.cat)

The human genome is constantly attacked by endogenous and exogenous genotoxic agents, which leads to genome instability, cancer development, and aging[1]. Cells counteract mutagenic insults through a complex DNA damage response network[2]. The biomedical relevance of genome maintenance is illustrated by the severe clinical consequences of mutations in DNA repair genes[1]. Genes involved in homologous recombination (HR) repair pathway such as *BRCA1*, *BRCA2*, and *PALB2* are an important example. Mutations in these genes cause familial breast cancer and the cancer-prone disease Fanconi anemia (FA) in monoallelic and biallelic carriers, respectively[3]. Furthermore, the proteins encoded by many of these genes are crucial for the modulation of the response of cancer cells to chemotherapeutics, including cisplatin and poly (ADP-ribose) polymerase (PARP) inhibitors[4]. Therefore, the identification of additional components of this DNA repair pathway is of utmost biomedical importance. Here we observe that EDC4, besides its known role in processing-bodies (P-bodies), interacts with BRCA1 and is involved in HR-mediated DNA repair by regulating its end-resection step and that germline mutations in *EDC4* may confer increased risk of breast cancer. Taking together our results suggest that EDC4 is a functional phenocopy of BRCA1 that could be targeted in cancer therapeutics.

## Results

### EDC4 interacts with TOPBP1 and associates with BRCA1.
TOPBP1 is required for HR repair[5, 6] and interacts with BRCA1 and BRIP1 in response to DNA damage[7]. To uncover novel proteins potentially involved in DNA repair and cancer predisposition, we screened for TOPBP1 interactors using the yeast two-hybrid system. Seven TOPBP1 baits were defined based on Pfam-predicted domains and PONDR-predicted disordered regions[8], which covered the complete protein sequence. A central putative disordered region in TOPBP1 (amino acids 643–836) used as a bait identified interactions with the enhancer of mRNA decapping protein 4, EDC4 (NCBI reference sequence: NP_055144.3; aliases: GE1, HEDLS RCD8; Fig. 1a). Notably, EDC4 was previously found to be post-translationally modified in response to DNA damage in proteomic studies[9, 10]. Four independent preys supported the physical TOPBP1–EDC4 interaction which was further confirmed by endogenous co-immunoprecipitation assays (Fig. 1b) and by co-affinity purification assays (Fig. 1c, d).

EDC4 is known to function in the mRNA P-bodies within the cytoplasm[11]. However, western blot analyses of cellular subfractions (Fig. 1e) and confocal microscopy using green fluorescent protein (GFP)-tagged EDC4 (see below) demonstrated that it is also located in the nucleus and binds to chromatin. Nucleoplasm localization is also supported by independent studies[12]. EDC4 contains a WD40-repeat domain in its N-terminal region. This type of domain is involved in the coordination of multi-protein complex assembly, and therefore, we hypothesized that EDC4 can interact with other TOPBP1 partners. Accordingly, we found that BRCA1 co-immunoprecipitates with BRIP1, as expected, and with EDC4 (Fig. 1f). These results suggest that EDC4 binds with BRCA1, BRIP1, and TOPBP1 in a nuclear complex.

### EDC4 is involved in DNA damage response.
Cells deficient in downstream components of the FA/BRCA signaling pathway, such as BRCA1 and BRIP1, are hypersensitive to DNA interstrand cross-linking (ICL) drugs, including mitomycin C (MMC) or diepoxybutane (DEB). To investigate whether EDC4 is similarly involved in genome maintenance, we depleted EDC4 in HeLa cells by RNA interference (RNAi), using three independent

small interfering RNAs (siRNA) that reached depletion levels of 89–96% (Fig. 2a). EDC4 depletion rendered HeLa cells hypersensitive to ICL drugs in terms of cell survival (Fig. 2b), MMC-induced G2/M cell cycle arrest (Supplementary Fig. 1a), and DEB-induced chromosome fragility, especially chromatid-type aberrations and chromatid-type exchanges (Fig. 2c and Supplementary Fig. 1b), whereas FANCD2 monoubiquitination in response to MMC-induced stalled replication forks was not affected (Supplementary Fig. 1c, d). To corroborate these findings, we used CRISPR/Cas9-mediated assays to completely abrogate EDC4 expression (i.e., knockout (KO)) in HEK293T cells (Supplementary Fig. 2a–c). Cell survival experiments showed that complete loss of EDC4 leads to hypersensitivity to DEB (Fig. 2d), exacerbated DEB-induced G2/M cell cycle arrest, and DEB-induced chromosome fragility, determined by the flow cytometric micronucleus (MN) assay (Supplementary Fig. 2d, e). Therefore, depletion of EDC4 results in DNA damage sensitivity, phenocopying deficiencies of downstream FA/BRCA pathway components. Next, similarly to BRCA1[13] and BRIP1[14] relocation to DNA double-strand breaks (DSBs), GFP-tagged EDC4 was found to be quickly (<5 s) allocated to DSB in vivo, as observed in laser micro-irradiation confocal microscopy experiments in U2OS cells (Fig. 3b), which is consistent with the nuclear function of EDC4 in DNA damage response in addition to its role at the cytoplasmic P-bodies.

Moreover, it is reported that EDC4 is post-translational modified by ubiquitination in lysines 514 and 1157 and by phosphorylation in serine 741 in response to DNA damage[10] (Supplementary Fig. 6a). We produced mutations of these residues (Supplementary Fig. 6b) and we observed that cell lines with mutations K514R and K1157R in EDC4 are sensitive to DEB agent and this treatment induced chromosome fragility (increase in MN frequency) and G2 arrest (Supplementary Fig. 6c–e), while the S741A mutant does not show DEB sensitivity (Supplementary Fig. 6c). These results indicate that the ubiquitination of EDC4 in lysine K514 and K1157 is crucial for DNA damage resistance.

### Independent roles of EDC4 in mRNA decapping and DNA repair.
Since EDC4 involvement in mRNA decapping and maintenance of genome integrity occurs in different cellular compartments, and as a part of different multiprotein complexes, we hypothesized that these roles are functionally independent. Initially, we used RNAi (Supplementary Fig. 1e) to deplete DCP1a (decapping protein 1a), a physical interactor of EDC4 with an essential role in P-body formation and mRNA decapping[15]. Depletion of EDC4 or BRCA1, but not DCP1a, resulted in MMC hypersensitivity (Fig. 3c) and MMC-induced cell cycle G2/M arrest (Fig. 3d and Supplementary Fig. 8), indicating that the lack-of-function P-bodies does not lead to DNA repair deficiency. As the C-terminal part of EDC4 is essential for the formation of P-bodies[16], we next generated a series of *EDC4* deletion mutants (Fig. 3e), selected single-cell clones expressing physiological levels of the truncated proteins (Supplementary Fig. 2f), and studied their proficiency in DNA damage sensitivity (Fig. 3f) and P-body formation (Fig. 3g). GFP-tagged EDC4 with a C-terminal deletion (Δ6) does not form P-bodies, but maintains its repair function as it genetically complements DEB hypersensitivity of *EDC4* KO HEK293T cells (Fig. 3f, g). In contrast, EDC4 with deleted WD40 domain (Δ2) forms P-bodies but is not functional in DNA repair (Fig. 3f, g). Therefore, these separation-of-function deletion mutants provide further functional evidence that the dual roles of EDC4 are mechanistically independent.

### EDC4 regulates HR-mediated repair and works with BRCA1.
Given the association of EDC4 with the TOPBP1-BRCA1-BRIP1

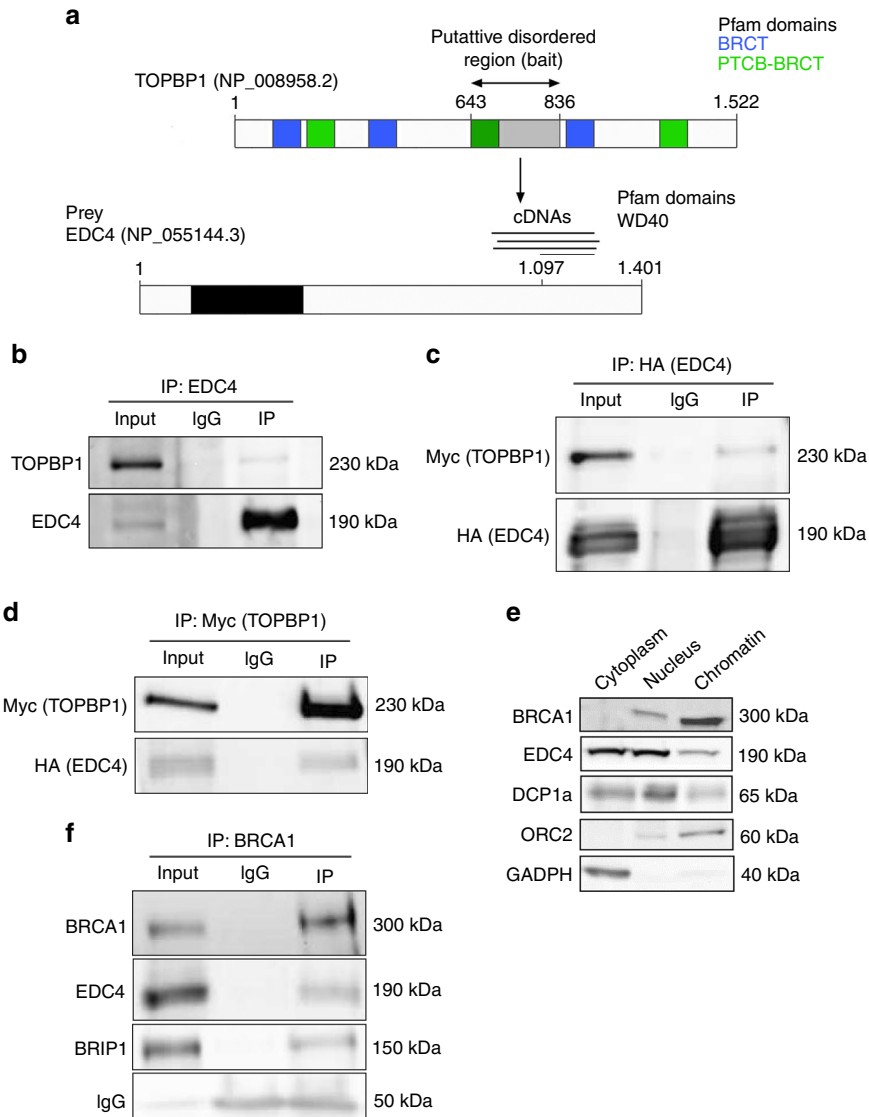

**Fig. 1** EDC4 interacts and with the BRCA1-BRIP1-TOPBP1 complex. **a** Diagram showing the region of TOPBP1 used as bait for the Y2H screen and the different cDNAs from *EDC4* captured. **b** Immunoblots showing that endogenous EDC4 interacts with TOPBP1 in HeLa cells. EDC4 was immunoprecipitated and analyzed by immunoblotting with indicated antibodies. **c** Immunoblots showing that exogenous EDC4 interacts with exogenous TOPBP1 in HeLa cells. EDC4 was immunoprecipitated from HeLa cells expressing both tagged EDC4 and TOPBP1 and analyzed by immunoblotting with indicated antibodies. **d** Immunoblots showing that exogenous TOPBP1 interacts with exogenous EDC4 in HeLa cells. TOPBP1 was immunoprecipitated from HeLa cells expressing both tagged EDC4 and TOPBP1 and analyzed by immunoblotting with indicated antibodies. **e** Cellular fractionation of HeLa cells shows that EDC4 is not only a cytoplasmatic protein but also present at the nucleus and the chromatin. **f** Immunoblots showing that endogenous EDC4 interacts with BRCA1 and BRIP1 in HeLa cells. BRCA1 was immunoprecipitated from HeLa cells and analyzed by immunoblotting with indicated antibodies

complex, we next assessed the involvement of EDC4 in HR-mediated repair. An in vivo HR assay was performed in U2OS cells by detecting HR-mediated restoration of an incomplete GFP expression cassette inserted in the genome[17] in the presence or absence of BRCA1, BRCA2, or EDC4. Control cells with intact HR repair pathway exhibited efficient functional GFP expression, while, as expected, BRCA1 and BRCA2 depletion by RNAi (Supplementary Fig. 3a) strongly abolished HR-mediated repair (Fig. 4a, b). Notably, EDC4 depletion resulted in a highly significant reduction in the percentage of fluorescent cells (Fig. 4a, b), indicating that EDC4 also participates in HR-mediated repair.

Since EDC4 interacts with BRCA1, and BRCA1 mediates DNA end resection at DSB, promoting error-free HR-mediated repair[18], we investigated whether EDC4 also regulates this process. Thus, reduced end-resection tracks were observed in

*EDC4* KO cells as measured by the single-molecule analysis of resection tracks (SMART) assay (Fig. 4c). Additionally, reduced formation of replication protein A (RPA) foci in EDC4-depleted cells was observed (Fig. 4d and Supplementary Fig. 3b). Moreover, since BRCA1[19] and BRCA2[20] are required for downstream loading of the RAD51 recombinase, we investigated MMC-induced formation of RAD51 foci in the absence of EDC4. The results demonstrated that EDC4 is also required for the formation of these foci (Fig. 4e and Supplementary Fig. 3b). Finally, we investigated whether EDC4-depleted cells are hypersensitive to PARP inhibition, another hallmark of HR deficiency[21, 22]. As shown in Fig. 4f and Supplementary Fig. 3c, EDC4 depletion renders HeLa cells hypersensitive to the PARP inhibitor veliparib.

Considering the fact that EDC4 and BRCA1 share common phenotypes, we studied the possibility that EDC4 and BRCA1

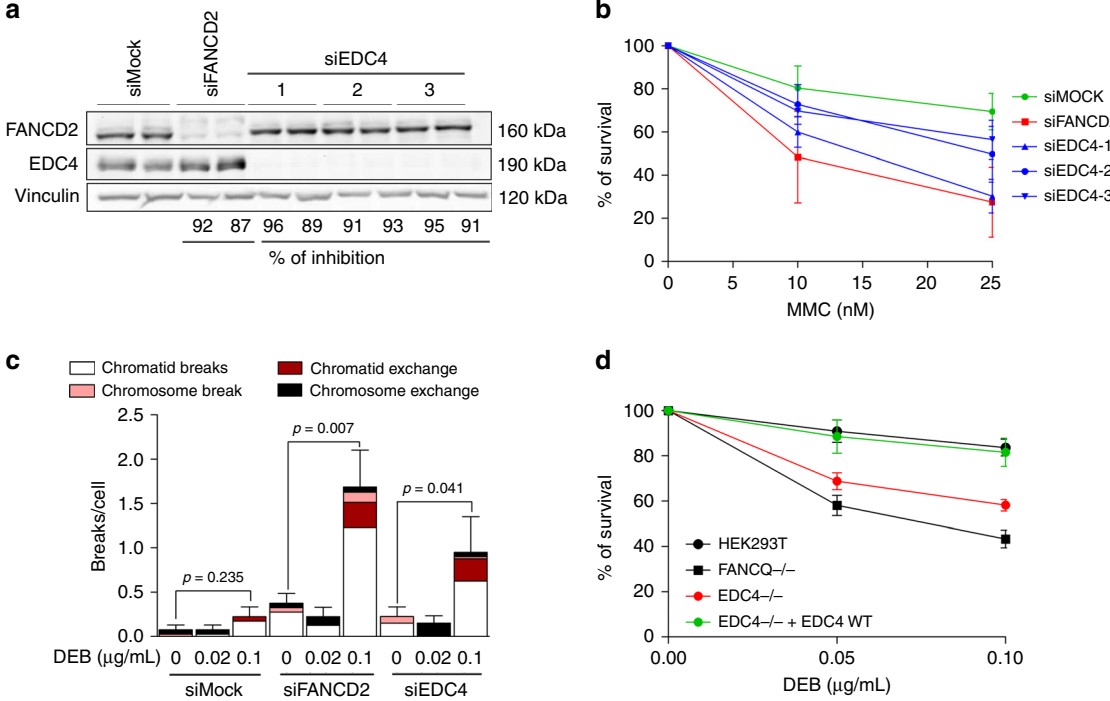

**Fig. 2** EDC4 is required for genome maintenance. **a** Immunoblots showing the inhibition efficiency in HeLa cells of FANCD2 and EDC4 by three different siRNA sequences used. Two replicas are shown for inhibition. **b** Inhibition of EDC4 reduces the survival of HeLa cells exposed to MMC. Data shown represent results from five independent experiments. Error bars indicate mean ± s.d. Means were statistically compared using the two-tailed Student's $t$ test. Statistical analysis comparing the means of the Mock-treated vs. siRNA-treated samples were performed with the following $p$ values: siFANCD2 (10 nM, $p = 0.0376$; 25 nM, $p = 0.0114$), siEDC4-1 (10 nM, $p = 0.0747$; 25 nM, $p = 0.0018$), siEDC4-2 (10 nM, $p = 0.0227$; 25 nM, $p = 0.0199$), and siEDC4-3 (10 nM, $p = 0.2004$; 25 nM, $p = 0.0028$). **c** EDC4 depletion causes chromosome fragility after DEB exposure in primary fibroblasts. Data shown represent results from the analysis of 40 metaphases (two experiments of 20 metaphases each one). Error bars indicate mean ± s.e.m. The statistical test performed in each inhibition to compare breaks per cell (0 vs. 0.1 μg/mL DEB) was a Mann–Whitney test. **d** CRISPR/Cas9-generated EDC4−/− HEK293T cell line shows a decrease in survival after DEB treatment that is reversed when the WT EDC4 is expressed. Data shown represent results from four combined independent experiments. Error bars indicate mean ± s.d. Means were statistically compared using the two-tailed Student's $t$ test. Statistical analysis comparing the means of the WT cell line with the samples were performed with the following $p$ values: FANCQ−/− ($p = 0.001$; $p = 0.001$), EDC4−/− ($p = 0.000$; $p = 0.000$), and EDC4 corrected ($p = 0.613$; $p = 0.173$)

work in the same pathway. We analyzed the effect of the simultaneous inhibition of EDC4 and BRCA1 in terms of DEB sensitivity and HR repair efficiency. Depletion of EDC4 and BRCA1 renders HeLa cells sensitive to MMC treatment in a similar way to the individual inhibition of EDC4 or BRCA1 (Fig. 4g and Supplementary Fig. 3d). We previously established that EDC4 shows a mild impairment in HR efficiency compared to BRCA1 (Fig. 4b). In this scenario, the inhibition of BRCA1 in cells inhibited for EDC4 shows impairment in repair similar to the sole inhibition of BRCA1 (Fig. 4h and Supplementary Fig. 3e).

Collectively, these data indicate that loss of EDC4 phenocopies BRCA1 deficiency, both being involved in the same pathway of genome maintenance by stimulating end resection at DSBs to promote error-free HR-mediated repair.

**Germline *EDC4* mutations in breast cancer patients**. Several downstream components of the FA/BRCA pathway have been associated with breast and/or ovarian cancer risk by high or moderate penetrance mutations[3]. Given the functional interaction of EDC4 with FA/BRCA components in HR-mediated repair, we investigated whether *EDC4* is also mutated in breast cancer cases. We initially sequenced the corresponding genomic coding region, comprising all exons and exon–intron boundaries, in blood DNA samples from 300 Spanish breast cancer index cases that fulfilled one of the criteria suggestive of hereditary breast and ovarian cancer (HBOC) syndrome, but were negative

for pathogenic mutations in *BRCA1* or *BRCA2* genes (see Methods). Targeted pooled DNA amplification followed by next-generation sequencing was used for mutation screening as previously described[23, 24]. After excluding common genetic variants, the analysis revealed five (1.67%) rare missense variants that were all confirmed by Sanger sequencing: c.125G>A, p.G42E; c.829A>C, p.S277R; c.1083C>A, p.D361E; c.1418G>A, p.R471Q; and c.1429G>A, p.V477M. Pedigrees are shown in Fig. 5a. These variants were present in 2 out of 736 (0.27%) Spanish controls (unrelated individuals not affected of neoplasms), according to the CIBERER Spanish Variant Server (odds ratio (OR) = 6,1, $p = 0.024$, Fisher's exact test). All variants were predicted to be deleterious by PolyPhen-2 HumVar (p.G42E, p.D361E, p.R471Q, and p.V477M) and/or CADD (p.S277R, p.R471Q, and p.V477M; Phred-like scores >20) algorithms. In addition, all variants were found to be clustered inside or near the functionally important WD40 domain of EDC4 (Fig. 5b) and in regions highly conserved in evolution (Supplementary Fig. 4). Inspection of the ExAc database that includes approximately 120,000 alleles did not reveal three of the above variants (G42E, S277R, and R471Q), and D361E and V477M were found 1 and 81 (2 homozygous) times, respectively. Thus, the ExAc frequency of *EDC4* variants previously detected in breast cancer cases was of 0.14%, which is lower than the CIBERER Spanish dataset. Noticeably, the ExAc study identified *EDC4* as an extremely intolerant gene to loss-of-function mutations[25]. In our study, evaluation of related

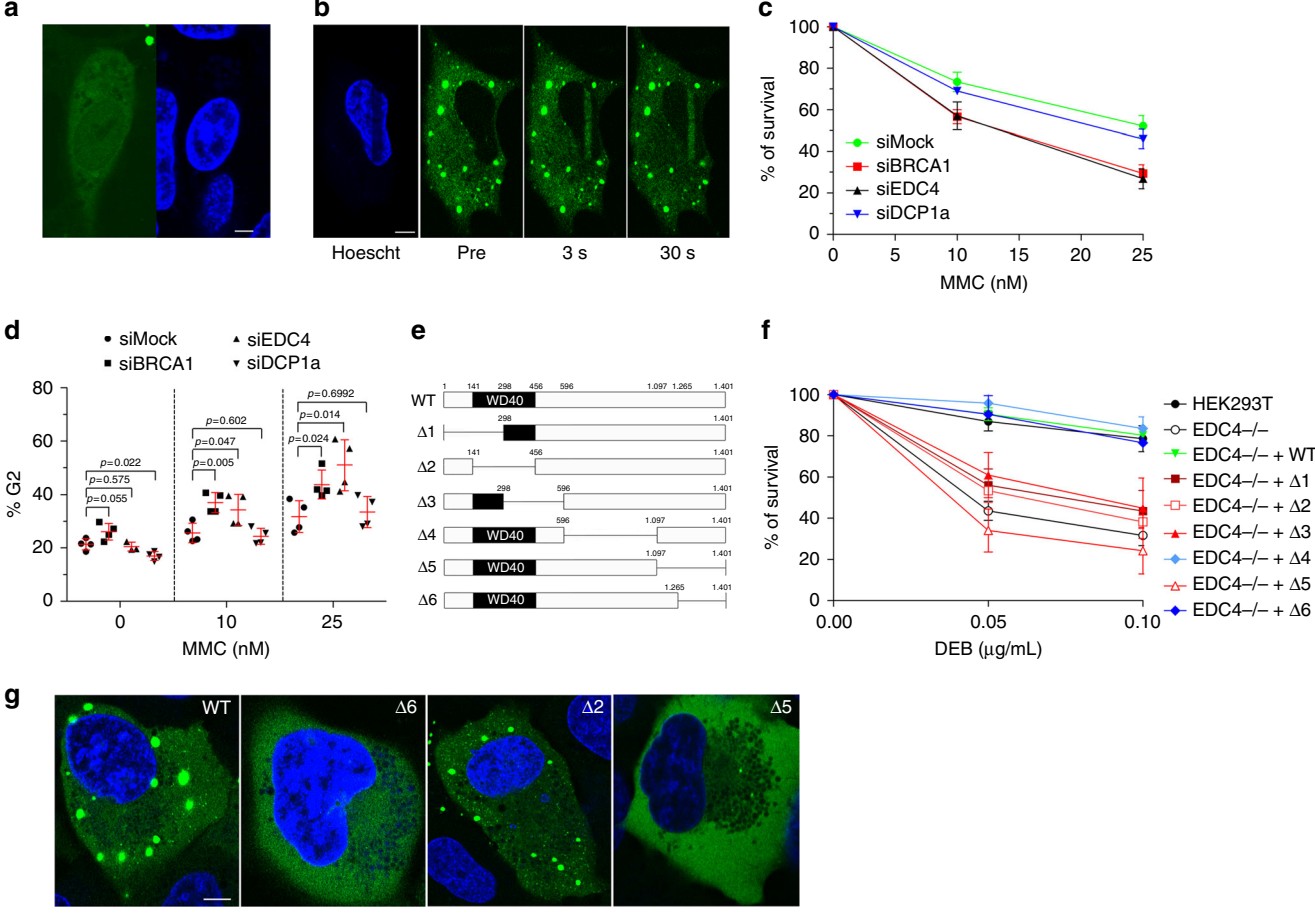

**Fig. 3** EDC4 is required for genome maintenance. **a** Confocal image of U2OS cells expressing GFP-tagged wild-type EDC4 showing an example of cell with nuclear localization of EDC4 in the absence of DNA damage. Scale bar represents 10 μm. **b** U2OS cells expressing GFP-EDC4 were subjected to laser micro-irradiation (see Methods). GFP-EDC4 images were taken before and after laser micro-irradiation by confocal microscopy. Scale bar represents 10 μm. **c**, **d** DCP1a is not required for the survival of HeLa cells exposed to MMC or for the G2/M blockade. Data shown represent results from four combined independent experiments. Error bars indicate mean ± s.d. Means were statistically compared using the two-tailed Student's *t* test. Statistical analysis comparing the means of the Mock-treated vs. gene-specific siRNA-treated samples for the sensitivity to MMC were performed with the following *p*values: siBRCA1 (10 nM, *p* = 0.0122; 25 nM, *p* = 0.0005), siEDC4 (10 nM, *p* = 0.0115; 25 nM, *p* = 0.0012), and siDCP1a (10 nM, *p* = 0.1574; 25 nM, *p* = 0.1862). **e** Diagram showing the regions of EDC4 deleted for functional studies. **f** Survival after DEB treatment of the six *EDC4* deletion mutants generated in HEK cells. Data shown represent results from at least two combined independent experiments. Error bars indicate mean ± s.d. Means were statistically compared using the two-tailed Student's *t* test. No statistically significant difference was observed between WT, Δ4 (*p* = 0.233; *p* = 0.516), Δ6 (*p* = 0.852; *p* = 0.455), and corrected cells (*p* = 0.8832; *p* = 0.4646), while the rest of the mutants were found statistically different compared with the WT cell line (EDC4−/−, *p* = 0.0013 and *p* = 0.0027; Δ1, *p* = 0.001 and *p* = 0.002; Δ2, *p* = 0.0009 and *p* = 0.0017; Δ3, *p* = 0.0542 and *p* = 0.0032; Δ5, *p* = 0.0001 and *p* < 0.0001). **g** Representative fluorescence images of U2OS cells expressing GFP-tagged WT EDC4 and deletion forms of GFP-EDC4 to demonstrate differences in P-body formation (cytoplasmic green aggregates). Scale bar represents 10 μm

individuals was limited and did not show co-segregation among the available affected relatives (Fig. 5a). However, analysis of a breast tumor in a V477M carrier showed loss of heterozygosity (Fig. 5a).

Subsequently, we investigated whether the identified rare missense variants were deleterious by disrupting the newly depicted EDC4 function in DNA damage repair. Thus, all five missense changes were introduced in an *EDC4* cDNA by site-directed mutagenesis in a lentiviral vector, subsequently transduced into *EDC4* KO cells, and evaluated in cultures derived from single-cell clones expressing physiological levels of each mutant EDC4 (Supplementary Fig. 5a). Importantly, none of the mutants complemented ICL-induced hypersensitivity (Fig. 5c), cell cycle arrest (Fig. 5d), or chromosome fragility (Fig. 5e) of *EDC4* KO cells, indicating that all mutations found in breast cancer patients disrupt the function of EDC4 in DNA repair. We then generated GFP fusions of the patient-derived missense mutations and

observed that they do not impair P-body formation (Supplementary Fig. 5b).

Given these results, we further explored data from whole-exome sequences of 50 Italian and 77 Spanish *BRCA1/2*-mutation-negative familial breast cancer cases and found an additional rare missense variant, c.440A>G p.Y147C in an Italian woman. Again, this variant was not reported in ExAc, it is predicted to be deleterious according to PolyPhen-2 and, furthermore, corresponds to a highly conserved residue within the WD40 domain (Supplementary Fig. 4). We genotyped this variant and found no carriers in 66 additional *BRCA1/2*-mutation-negative cases and 702 healthy controls (female blood donors) from Italy, which is compatible with the hypothesis that the Y147C is a rare deleterious mutation. Unfortunately, the original breast cancer mutation carrier died, and no additional studies of the family members were possible. Finally, several breast cancer susceptibility genes with a role in HR-mediated

repair are found to be mutated in FA patients. To investigate whether *EDC4* is also mutated in FA, we studied DNA or cell lines from 17 FA patients with normal FANCD2 ubiquitination and without mutations in any of the 22 known FA genes[26]. We did not find *EDC4* mutations in 15 patients' DNAs or lack of EDC4 expression in 12 FA cell lines (Supplementary Table 2 and Supplementary Fig. 7). While additional genetic analysis in larger cohorts may be needed, our data suggest that rare variants in

*EDC4* may influence breast cancer susceptibility but lack a connection to FA.

## Discussion

This study demonstrates that EDC4 has two independent functions: decapping of mRNA at the P-bodies in the cytoplasm[11], and HR-mediated DNA damage repair in the nucleus. Critically,

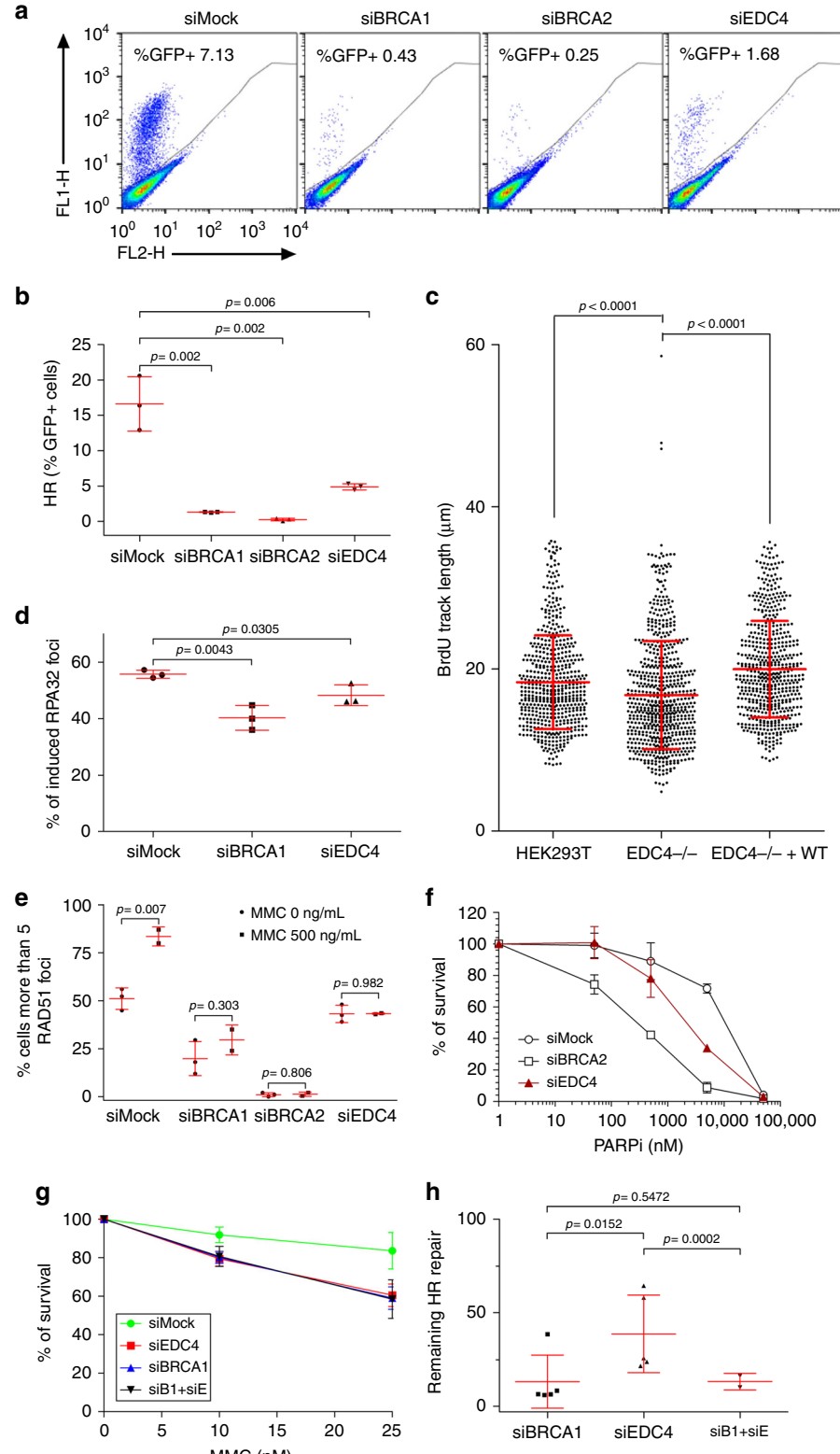

loss of EDC4 phenocopies the alteration of the FA/BRCA signaling pathway and of BRCA1 loss of function and both are involved in the same pathway. A number of functional connections between RNA metabolism and DNA repair exist, such as transcription-coupled nucleotide excision repair or R-loop-related prevention of genome instability[27], with an important role of several FA/BRCA pathway components[28–31]. It is also known that several proteins involved in the smallRNA machinery like DROSHA and DICER affect the recruitment of certain DNA damage response factors like MDC1 and 53BP1[32], thus reinforcing the idea of an interplay between DNA repair and RNA metabolism. However, to the best of our knowledge, EDC4 is the first protein shown to have a dual role in DNA repair and mRNA decapping. Given the dual role of EDC4, it would be interesting to investigate whether EDC4 plays a role in homology-directed repair using RNA as a template for DSB repair in a way similar to Rad52[33]. Our results indicate that all mutations identified in breast cancer cases are inside or near the WD40 domain of EDC4 at the N terminus of the protein. Supporting the importance of an intact WD40 domain, large deletions involving all or part of the WD40 domain specifically disrupted the function of EDC4 in DNA repair (Fig. 3e, f), indicating that this is one of the most relevant regions for EDC4 functions in genome maintenance and, likely, breast cancer protection. However, the region 1097–1265 contained in the C-terminal deletion 5 (Δ5) also has a role in DNA damage resistance (Fig. 3e, f). Corroborating the need of both the N terminal and C terminal for DNA repair is the fact that the Δ4 deletion is able to rescue the DNA damage sensitivity of EDC4 deficiency (Fig. 3e, f).

Consistent with a function in DNA repair, EDC4 was reported to be post-translationally modified by phosphorylation (S741) or ubiquitination (K514 and K1157) in response to DNA damage in proteomic screens[9, 10]. We produced mutations of these three residues (S741A, K514R, and K1157R) and detected that K514 and K1157, but not S741, are essential for EDC4 functions in DNA damage resistance (Supplementary Fig. 6c). In accordance with these results, our battery of EDC4 deletion mutants indicated that the regions comprising the ubiquitination target K514 (Δ3) or K1157 (Δ5), but not the phosphorylation target S741 (Δ4), are crucial for DNA damage resistance (Fig. 3e, f). We then generated GFP fusions of these ubiquitination mutants and observed that they do not impair P-body formation (Supplementary Fig. 5b). Study of the role of ubiquitination in EDC4 regulation and identification of the responsible ubiquitin ligase may further decipher the role of this protein in DNA repair and its link to cancer susceptibility. Given that EDC4 interacts with BRCA1, it is probably a member of at least one of the multiple BRCA1-containing protein complexes: BRCA1-A, BRCA1-B, BRCA1-C, BRCA1/PALB2/BRCA2, and the BRCA1/BARD1 heterodimer[34]. Each of these complexes has different functions related to DNA repair and cell cycle checkpoint activation and some of these complexes have antagonistic functions. BRCA1-B and BRCA1-C are known to promote DSB end resection[18], thus promoting HR, but BRCA1-A limits it resulting in a reduction in HR repair[35].

Our present results indicate that EDC4 mutation carriers are more frequent in breast cancer cases than in controls, and, therefore, we propose to include EDC4 in larger sequencing studies. In parallel, given the role of EDC4 in HR repair and the hypersensitivity of EDC4-depleted cell lines to PARP inhibition (Fig. 4f), our results suggest a therapeutic option for the treatment of EDC4-mutated breast cancers by synthetic lethality. It remains to be determined the involvement of this gene/protein in different carcinogenic processes in which mutations of FA/BRCA DNA repair genes are found relatively frequent, including ovarian, pancreatic, and metastatic prostate cancer[4, 36]. Interestingly, EDC4 has also been identified as frequently mutated in metastatic breast cancer, in parallel to PALB2 and other genes[37]. Moreover, whole-exome sequencing data from 412 high-grade serous ovarian cancer patients from The Cancer Genome Atlas Project identified two cases with germline truncating mutations in the N-terminal region of EDC4 (c.508_509delAC and c.689_690delAC)[38]. Therefore, independent observations further suggest a role for altered EDC4 function in cancer development and/or progression. In conclusion, our study demonstrates that EDC4 functionally phenocopies BRCA1, playing a role in HR-mediated DNA repair by regulation of end resection, and that germline mutations in EDC4 may confer risk of breast cancer. Further studies are required to determine its role as tumor suppressor and its potential use as molecular target in cancer therapeutics.

## Methods

**Immunoprecipitation.** HeLa cells seeded in 15 cm diameter petri dishes were trypsinized and recovered. Cell pellet was washed twice with cold phosphate-buffered saline (PBS) and cells were lysed with NTEN buffer (NP-40 0.5%, Tris 20 mM, pH 7.4, NaCl 150 mM, EDTA 1 mM, pH 8, and benzonase) for 1 h at 4 °C in rotation. Cell extracts were centrifuged at $2900 \times g$ at 4 °C for 10 min and the supernatant was recovered. Protein extracts were precleared for 1 h at 4 °C in rotation using magnetic beads (Protein G Mag Sepharose Xtra, 28-9670-66 GE Healthcare) and unspecific IgG in a proportion of 1 μg of antibody per 1 mg of protein extract, followed with a second preclearing step using only magnetic beads for 1 h at 4 °C in rotation. After that, protein extracts were separated in tubes and incubated with specific and unspecific antibodies and magnetic beads overnight at 4 °C in rotation. The beads were washed four times with NTEN buffer and the proteins were recovered with Laemmli buffer and analyzed by western blot.

**Cell fractionation.** Cells were harvested and washed with PBS once. The cellular pellet was then resuspended in buffer A (0.1 Triton X-100, HEPES 10 mM, pH 7.9,

**Fig. 4** EDC4 is involved in homologous recombination repair and works in the same pathway as BRCA1. **a** %GFP+ cells after I-SceI expression in siRNA-silenced Mock, BRCA1, BRCA2, and EDC4 U2OS-DR-GFP cells. **b** Quantification of the HR proficiency of BRCA1, BRCA2, and EDC4 siRNA-silenced U2OS-DR-GFP cells. Data represent results from three experiments. Error bars indicate mean ± s.d. Means were statistically compared using the two-tailed Student's t test. **c** Results of three SMART experiments in HEK293T cells. A significant reduction in resection track length is observed in EDC4−/−, but not in genetically complemented EDC4−/− cells. Data shown represent results from three experiments. Error bars indicate mean ± s.d. Statistical analysis was the Mann–whitney test. **d** EDC4 is required for normal RPA32 foci formation after 4 h MMC (500 ng/mL) treatment in HeLa cells. Data shown represent results from three experiments. Error bars indicate mean ± s.d. **e** EDC4 is required for normal loading of RAD51 onto DSBs after 4 h MMC (500 ng/mL) treatment in HeLa cells. Data shown represent results from at least two experiments. Error bars indicate mean ± s.d. Means were statistically compared using the two-tailed Student's t test. **f** EDC4 depletion renders HeLa cells hypersensitive to Veliparib after a 3-day treatment. Data shown represent results from three experiments. Error bars indicate mean ± s.d. Means were statistically compared to the Mock using two-tailed Student's t test: siBRCA2 ($p = 0.0124$; $p = 0.0023$; $p < 0.0001$; $p = 0.0815$), siEDC4 ($p = 0.8183$; $p = 0.3206$; $p < 0.0001$; $p = 0.2856$). **g** Simultaneous inhibition of BRCA1 and EDC4 in HeLa cells behave similarly to the single gene inhibition after DEB treatment. Data shown represent results from three experiments. Error bars indicate mean ± s.d. Means were statistically compared to the Mock using two-tailed Student's t test: siEDC4 ($p = 0.0181$; $p = 0.0231$), siBRCA1 ($p = 0.0158$; $p = 0.0188$), and siBRCA1-siEDC4 ($p = 0.0423$; $p = 0.0341$). **h** Quantification of the remaining HR repair of BRCA1, EDC4, and BRCA1/EDC4 siRNA-silenced U2OS-DR-GFP cells. Data shown represent results from at least two experiments. Error bars indicate mean ± s.d

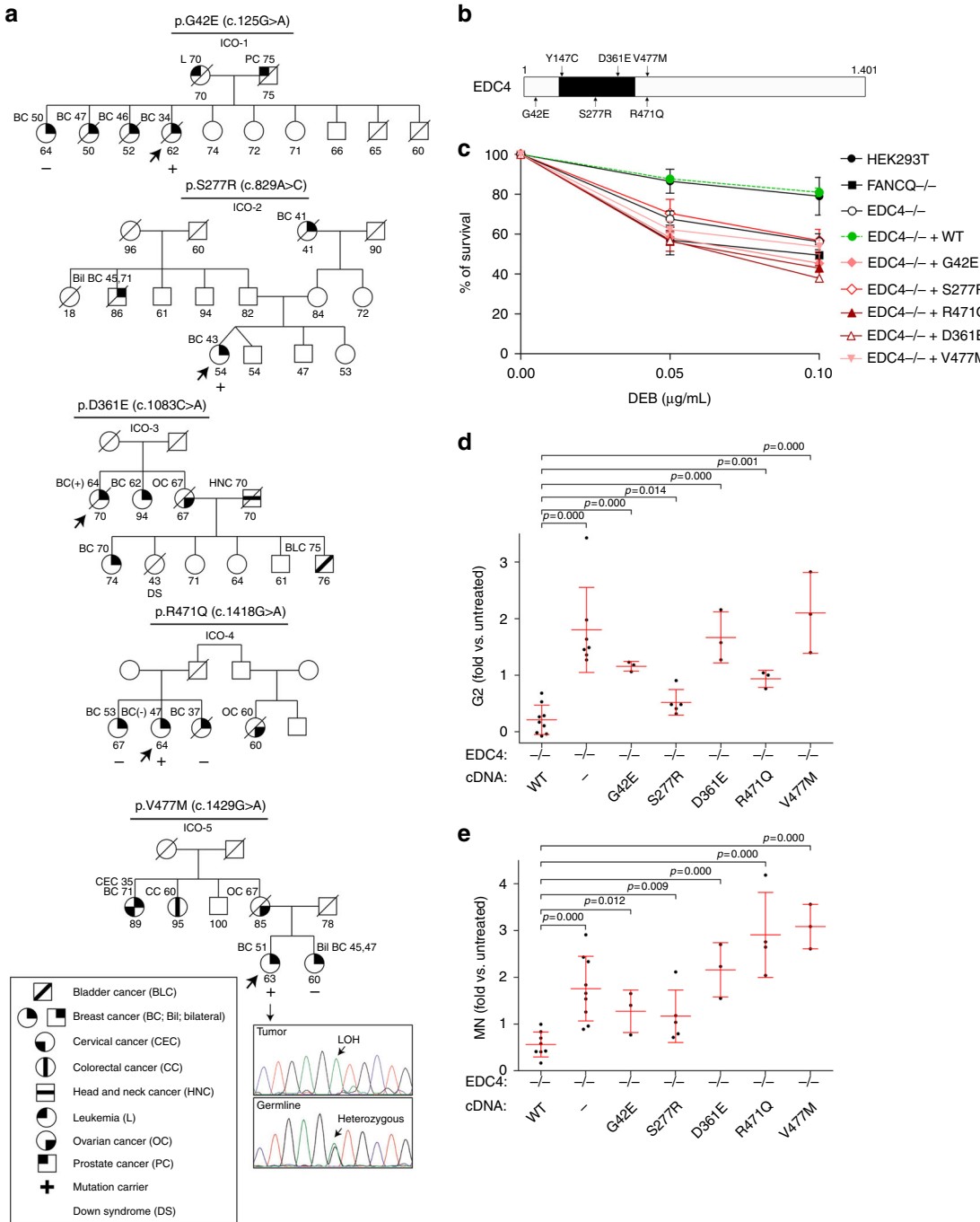

**Fig. 5** Rare *EDC4* germline variants identified in breast cancer patients are functionally deleterious. **a** Pedigrees of the five breast cancer families with *EDC4* mutations (based on NP_055144). The probands are indicated by arrows, and slashed symbols denote deceased individuals. The ages at diagnosis and/or death are included when known. Individuals genotyped for mutations are marked either as carriers (+) or non-carriers (−). Additional clinical annotations are depicted as shown in the inset. **b** Diagram showing that all the residues mutated in *BRCA1/2*-mutation-negative patients cluster inside or close to the WD40 domain of EDC4. **c** Functional studies of mutations found in patients shown in **a**. All *EDC4* mutants fail to revert the DEB sensitivity phenotype of HEK293T *EDC4−/−* cells. Graph shows survival after DEB treatment of five *EDC4* mutants generated in HEK293T cells. Data shown represent results from at least two combined independent experiments. Error bars indicate mean ± s.d. Means were statistically compared using the two-tailed Student's *t* test: EDC4−/− (p = 0.000; p = 0.000), EDC4 corrected (p = 0.760; p = 0.748), G42E (p = 0.002; p = 0.005), S277R (p = 0.024; p = 0.011), D361E (p = 0.001; p = 0.001), R471Q (p = 0.010; p = 0.017), and V477M (p = 0.000; p = 0.000). **d** HEK293T *EDC4−/−* cells expressing the mutants show increased G2/M block induced by DEB treatment. **e** HEK293T *EDC4−/−* cells expressing the mutants show DEB-induced chromosome fragility as shown with the flow cytometric MN assay. Data shown in **d**, **e** represent results from at least three combined independent experiments. Error bars indicate mean ± s.d. Means were statistically compared using one-way analysis of variance (ANOVA) followed by a Dunnett's multiple comparison test

KCl 10 mM, $MgCl_2$ 1.5 mM, sucrose 0.34 M, glycerol 10%, DTT (dithiothreitol) 1 mM) and left 8 min in ice. Suspension was then centrifuged 5 min at $1300 \times g$ at 4 °C. The supernatant, containing the cytoplasmic fraction, was transferred to a new tube. The nuclear pellet was washed once with buffer A and centrifuged 5 min at $1300 \times g$ at 4 °C. The nuclear pellet was then resuspended in RIPA buffer.

**Immunoblot analysis.** Cells were trypsinized and then lysed for 10 min at room temperature in RIPA lysis buffer (0.5 M Tris-HCl, pH 7.4, 1.5 M NaCl, 2.5% deoxycholic acid, 10% NP-40, 10 mM EDTA) supplemented with protease and phosphatase inhibitors and benzonase. Forty to eighty micrograms of total proteins were separated on 6–8% sodium dodecyl sulfate–polyacrylamide gel electrophoresis (SDS-PAGE) gel, blotted onto nitrocellulose membranes, and incubated overnight with the appropriate primary antibody in 5% bovine serum albumin in TTBS (20 mM Tris, 0.5 M NaCl, pH 7.5, and 0.1% Tween-20). Blots were then detected with horseradish peroxidase (HRP)-conjugated secondary antibodies. Finally, proteins were visualized with Luminata Crescendo Western HRP substrate (Merck Millipore) or Pierce ECL Western Blotting substrate (Thermo Scientific). Uncropped scans of the most important blots are shown in Supplementary Figs. 9–12.

**FANCD2 monoubiquitination analysis.** HeLa cells were transfected with specific siRNA and treated with 2 mM hydroxyurea for 24 h. Protein whole cell extracts were prepared using RIPA buffer as previously described. Forty micrograms of total proteins were separated on 6% SDS-PAGE gel, blotted onto nitrocellulose membrane, and incubated overnight with anti-FANCD2 primary antibody in 5% bovine serum albumin (BSA) in TTBS (20 mM Tris, 0.5 M NaCl, pH 7.5, and 0.1% Tween). Finally, blots were detected with anti-rabbit HRP-conjugated secondary antibody. Both bands of FANCD2 were quantified using ImageJ program and the result was expressed as an ub-FANCD2/nonub-FANCD2 ratio.

**MMC survival and G2 arrest after siRNA-mediated inhibitions.** HeLa cells were transfected with specific siRNA. After that, cells were untreated or treated with 10 and 25 nM of MMC for 72 h, medium was discarded, and cells were trypsinized. Cell number and cell cycle was simultaneously analyzed by cell cytometry using Perfect-count microspheres (CYT-PCM-100, Cytognos) and propidium iodide (Invitrogen). Results for survival were expressed in percentage related to untreated controls. The percentage of G2 cells was calculated from the cytometer data using the FlowJo program. When the intensity of the propidium iodide was evaluated in a histogram, the G2 pick was selected and the percentage of cells in G2 was calculated related to the total amount of cells that were evaluated for cell cycle distribution.

**Chromosome fragility assay.** Chromosome fragility test was done on the basis of diepoxybutane-induced chromosome fragility tests, as previously described[39, 40]. Briefly, wild-type (WT) primary fibroblast were treated with specific siRNA and seeded in culture flask. Cells were then treated with diepoxybutane (0.02 and 0.1 μg/mL) for 4 days. Four hours before fixation cultures were treated with colcemid (Gibco). Then cultures were fixed in methanol:acetic acid and metaphase spreads were prepared and stained with Giemsa using standard cytogenetic techniques. Metaphase cytogenetic analysis was performed in a Zeiss AXIO Imager M1 microscope coupled to a computer-assisted metaphase finder (Metasystems, Germany).

**Diepoxybutane survival.** HEK293T cells were seeded in 6-well plates (150,000 cells per well) and 24 h later were treated with 0.05 and 0.1 μg/mL of diepoxybutane (Sigma) for three population doublings. After that, cells were harvested, and the cell number was assessed using the Beckman Cell Counter. The results were expressed as the percentage of survival relative to untreated cultures.

**Cell cultures and plasmids.** U2OS cells were obtained from the American Type Culture Collection, HEK293T cells were kindly provided by Joan Seoane of Hospital Vall d'Hebron (Barcelona, Spain). HeLa cells were kindly supplied by Maria Blasco of Spanish National Cancer Research Center (CNIO) (Madrid, Spain). U2OS-DR-GFP cells were kindly provided by Maria Jasin of Memorial Sloan Kettering Cancer Center (New York, USA). All cell lines were tested for mycoplasma contamination.

U2OS, HeLa, HEK293T, and stable cell line U2OS-DR-GFP were grown in Dulbecco's modified Eagle's medium (DMEM) (Biowest, cat. no. L0104) supplemented with 10% fetal bovine serum (Biowest, cat. no. S181B) and plasmocin 0.1 mg/L (Invivogen, cod ant-mpt). I-SCEI-expressing plasmid pCBAS, empty vector (pCAGGS), and GFP-expressing plasmid NZE-GFP were kindly provided by Dr. Maria Jasin (Memorial Sloan Kettering Cancer Center, New York). pRetroQ-GFP-EDC4-transfected U2OS cells were sorted twice (FacsJazz, BD Biosciences) for stable transfection and selection maintained with puromycin 0.5 μg/mL. For DNA and siRNA transfection, we used Lipofectamine 2000 (cat. no. 11668) and Lipofectamine RNAiMax (cat. no. 13778), respectively, from Invitrogen.

**Laser micro-irradiation experiments.** Cells were plated, stimulated with 10 μM 5-bromo-2-deoxyuridine (BrdU) for 24 h, and before the microscopic analysis they were pre-treated for 10 min at 37 °C with 5 μg/mL Hoescht 33342 (Life Technologies). Images were taken with an Olympus Fluoview 1000 confocal microscope. For laser micro-irradiation studies, cells were microirradiated with 1-s pulse five times covering the area with the 405-nm laser at full power and images taken every 3 s for up to 2 min after laser micro-irradiation.

**Plasmid transfection.** Cells were transfected with plasmids using Lipofectamine 2000 (Invitrogen, 11668-019) according to the manufacturer's instructions. Briefly, a dilution of plasmid and Lipofectamine was made using OPTI-MEM I 1× (GIBCO, cat. no. 31985-047) and then added to the cells. After 4 h, the medium was removed and fresh complete DMEM medium was added. Cells were allowed to grow for 48 h before analysis.

**Plasmid construction.** The pCMV6-EDC4-HA-HIS was purchased from OriGene (PS100008). The fragment containing EDC4-HA-HIS was extracted by digesting with AsiSI and PmeI and then cloned into pGK lentiviral vector modified to contain AsiSI and PmeI target sequences. To generate the EDC4 mutant forms, QuikChange II XL Site-Directed Mutagenesis Kit from Agilent Technologies (200521) was used with specific primers for each mutant (Supplementary Table 1). For generation of EDC4-Δ4 mutant, a PCR-ligation approach was used. Briefly, using specific primers (Supplementary Table 1) two fragments were amplified. Fragments 1 and 2 corresponding to the 5′ and 3′ side, respectively, of the point of deletion. The reverse primer from fragment 1 and the forward from fragment 2 are overlapping. In a second PCR, both fragments are mixed and overlapping regions work as primers for the final amplification generating the EDC4-Δ4 construct. Construct was then digested with AsiSI and PmeI and finally cloned into pGK lentiviral vector. For pRetroQ-GFP-EDC4 construct, EDC4 was obtained from pCMV6-EDC4-HA-H6 by PvuI+PmeI digestion, blunt ended, and cloned into SalI-digested and blunt-ended pRetroQ-AcGFP1-C1 vector (Clontech).

**CRISPR-Cas9 KO generation.** The CRISPR-sgRNA (single guide RNA) construct was generated ligating the sgRNA sequence targeting the exon 5 of EDC4 (5′-TGGTGCGGGTGATCAGCGTC-3′) onto the pX330 plasmid containing the Cas9 gene (Addgene 42230[41]). This target sequence was bioinformatically designed to minimize off-target effects[41]. To assess the specificity of the CRISPR endonuclease activity, a fluorescent reporter construct was generated ligating the pRG2S vector (Labomics[42]) with the same sgRNA sequence used previously. Briefly, HEK293T cells were transfected with both the reporter and the CRISPR constructs. Three days after transfection, RFP$^+$GFP$^+$ cells were sorted using the FACSAria II (BD Biosciences) and subsequently seeded at one cell per well in 96-well plates. Clones were analyzed for EDC4 protein levels and those without expression of the protein were genotyped.

**Genotyping of EDC4 KO clones.** Genomic DNA was extracted using DNeasy Blood and Tissue Kit (Qiagen) following the manufacturer's instructions. The region surrounding the designed CRISPR target site was amplified using the following primers 5′-TGTTCTGCTGGATGTCCCAC-3′ and 5′-TCTCCTCAGG-GATGAAGGGG-3′. PCR amplification products were purified using ExoSAP-IT (Affymetrix) following the manufacturer's instructions and sequenced using Sanger method in order to identify homozygous mutant clones.

**Generation of lentiviral particles.** For lentiviral particle production, the CalPhos Mammalian Transfection Kit (Clontech, 631312) was used to transfect the following plasmids onto early passages of HEK293T cells: EDC4-WT or mutant forms, the packaging plasmid (PAX), and the lentiviral VSV-G (ENV)[43]. Two 10-cm petri dishes were seeded with $5 \times 10^6$ cells for each virus. Before transfection, the culture medium was changed and 3 μL of chloroquine 100 mM were added. Then, for each dish, 10 μg of the EDC4-WT or mutant plasmid were mixed with 6.5 μg of PAX plasmid, 3.5 μg of ENV plasmid, 87 μL of $CaCl_2$, and 593 μL of $H_2O$. While gently mixing with the vortex, 700 μL of 2× HBS solution was added dropwise. After 15 min of incubation, 1.4 mL of the solution was added to each plate, and 24 h later, the medium was changed. The medium containing the lentiviral particles was collected 48 and 72 h after transfection, then filtered using a 0.42 μm filter and concentrated by centrifugation at 4000 rpm for an hour using Amicon Ultra-15 filtering units (Millipore). The concentrated lentiviral particles were kept at −80 °C. For viral infection, HEK293T EDC4−/− cells were seeded in a 12-well plate and infected using 50 μL of lentiviral particles. Cells were allowed to grow for 3 days before seeding them at one cell per well in 96-well plates. Clones were selected for similar expression levels of EDC4 compared with the HEK293T WT cell line.

**HR assay.** This method was previously described[17]. Briefly, the U2OS cell line stably transfected with one single copy of DR-GFP construct was inhibited with specific siRNAs. Then, cells were transfected with an I-SceI endonuclease expression vector (pCBASce) or with an empty vector (pCAGGS) or with a plasmid that expresses GFP constitutively (pNZE-GFP). After 48 h post transfection, cells were

processed to be analyzed by flow cytometry. The percentage of HR was quantified as percentage of GFP+ cells after I-SceI transfection once they were corrected by the efficiency of transfection and by cell cycle distribution.

**End-resection assay**. SMART was performed as described[18]. Briefly, cells were grown in the presence of 10 µM BrdU for 24 h. Cultures were then irradiated (10 Gy) and harvested after 1 h. Cells were embedded in low-melting agarose (Bio-Rad), followed by DNA extraction. DNA fibers were stretched on silanized coverslips, and an immunofluorescence was carried out to detect BrdU. Samples were observed with a Nikon NI-E microscope, and images were taken and processed with the NIS ELEMENTS Nikon Software. For each experiment, at least 200 DNA fibers were analyzed, and fiber length was measured with Adobe Photoshop CS4.

**Immunofluorescence and foci formation**. Immunofluorescence protocol was previously described[44] and performed with some modifications. Briefly, cells were seeded in coverslips. Cells were then fixed with 4% paraformaldehyde and permeabilized with PBS containing 0.5% Triton X-100. Cells were then blocked with PBS containing 5% BSA and 0.05% Tween-20. Coverslips were incubated with primary antibodies and then with secondary antibodies conjugated to Alexa Fluor 488 or Alexa Fluor 568. Finally, samples were stained with DAPI (4′, 6-diamidino-2-phenylindole). Microscopic analysis and quantification were performed using a Zeiss AXIO Observer fluorescence microscope. When RAD51 foci formation was analyzed, cells were treated with 500 ng/mL MMC and left growing for 5 h. Thirty minutes before fixation the cells were labeled with 10 µM of 5-ethynyl-2′-deoxyuridine (EdU). Then, cells were subsequently fixed and processed following the manufacturer's instructions with the Click-it EdU Alexa Fluor 488 Imaging Kit (C10337, Invitrogen). Rad51 foci quantification was done by analyzing 200 EdU+ cells that were classified according to the number of cells with Rad51 foci (≤ or >5 foci per cell). For RPA32 foci formation, cells were treated with 500 ng/mL MMC for 2 h and processed as previously described. RPA32 foci quantification was done by analyzing 200 EdU+ cells classified according to the number of cells with RPA32 foci (≤ or >5 foci per cell).

**PARP inhibitors**. HeLa cells were plated in six-well petri dishes at a density of 1 × 10^5 cells per well. After 24 h they were untreated or treated with 50, 500, 5000, or 50,000 nM of Veliparib (ABT-888, Selleckchem) and kept in culture for 72 h. Cell number was analyzed using Beckman Cell Counter. Results were expressed as a percentage of survival compared to untreated cultures.

**RNA interference**. Cells were transfected with siRNA using Lipofectamine RNAiMAX (Invitrogen cat. no. 13778-150) according to the manufacturer's instructions. Briefly, a dilution of siRNA and Lipofectamine was made using OPTI-MEM I 1× (GIBCO, cat. no. 31985-047) and then added to the cells. After 4 h, the medium was removed and fresh complete DMEM medium was added. Cells were allowed to grow for 24 h before repeating the siRNA transfection. Forty-eight hours after the second transfection, inhibition levels were assessed using western blot. The following siRNAs were used: Luciferase (5′-CGUACGCGGAAUACUUCGA-3′), EDC4-1 (5′-CAUAUCACCUGCUGCAGCA-3′), EDC4-2 (5′-CAGGAAUA-CUUGCAGCAGCUA-3′), EDC4-3 (5′-CACUGAAGGCCAGCAGACAG-3′), FANCD2 pool (5′-CCUCGACUCAUUGUCAGUCAACUAA-3′/5′-CCAUGU-CUGCUAAAGAGCGUUCAUU-3′/5′-GGUGAUGGAUAAGUUGUCGU-CUAUU-3′), BRCA1 (5′-GUGGGUGUUGGACAGUGUA-3′), BRCA2 (5′-GGAUUAUACAUAUUUCGCA-3′), and DCP1a (SMARTpool J-021242-08, GE Healthcare).

**Antibodies**. The following antibodies were used: mouse anti-BRCA1 (OP92, Calbiochem) 1:2000, rabbit anti-BRCA2 (ab123491, Abcam) 1:2000, rabbit anti-DCP1a (ab47811, Abcam) 1:1000, rabbit anti-EDC4 (ab72408, Abcam) 1:1000, rabbit anti-FANCA (A301-980A, Bethyl Laboratories) 1:1000, rabbit anti-FANCD2 (ab2187, Abcam) 1:2500, rabbit anti-BRIP1 (ab16608, Abcam) 1:1000, rabbit anti-GAPDH (ab9485, Abcam) 1:1000, rabbit anti-HA tag (ab9110, Abcam) 1:1000, mouse anti-ORC2 (ab31930, Abcam) 1:1000, rabbit anti-RAD51 (8349, Santa Cruz) 1:250, rabbit anti-TOPBP1 (ab2402, Abcam) 1:1000, mouse anti-vinculin (ab18058, Abcam) 1:5000, goat anti-rabbit-Alexa 568 conjugated (A11036, Molecular Probes) 1:500, goat anti-rabbit-Alexa 488 conjugated (A11034, Molecular Probes) 1:500 and goat anti-mouse-Alexa 568 conjugated (A11031, Molecular Probes) 1:500, goat anti-rabbit IgG-HRP conjugated (A120-201P, Bethyl Laboratories) 1:1000, and goat anti-mouse IgG-HRP conjugated (sc-2005, Santa Cruz) 1:1000.

**Patients**. The 300 cases correspond to high-risk families with suspected Hereditary Breast and Ovarian Cancer Syndrome examined under the Hereditary Cancer Program of the Catalan Institute of Oncology (ICO) in Barcelona, Catalonia, Spain. In all cases, all relevant ethical regulations were applied. Informed consent was obtained from all the patients and the study was approved by the local Ethical Board, Bellvitge Biomedical Research Institute (IDIBELL IRB Board; PR138/16). The cases included in this study were negative for *BRCA1* or *BRCA2* mutations and matched one of the following criteria: at least three first-degree relatives affected by breast or ovarian cancer; or at least two first-degree female relatives affected by breast cancer (at least one of them diagnosed before the age of 50 years); or at least one case of female breast cancer plus at least one case of either ovarian, female bilateral breast, or male breast cancer. Genetic counselors collected clinical and pathological data from affected carriers.

**Mutation identification in pooled samples**. Index patients were screened for new mutations in *EDC4* by using a combination of pooled samples, PCR amplification, and high-throughput sequencing, as previously described[24]. Amplification was performed using Phusion High-Fidelity DNA Polymerase (New England Biolabs, Ipswich, MA, USA) and custom-designed primers. Normalized amounts of genomic DNA were used in PCR assays to amplify selected exons. Equimolecular amounts of each amplicon were pooled and random ligation of amplicons was performed in order to obtain concatemers, which were fragmented and subsequently used for library preparation by applying a Paired-End protocol (Illumina). Next-generation sequencing was performed on a HiSeq-2000 at the CNAG (Barcelona, Spain).

**Flow cytometric micronuclei assay and cell cycle analysis**. HEK293T WT, FANCA−/−, EDC4−/−, EDC4 corrected, EDC4 G42E, EDC4 S277R, EDC4 D361E, EDC4 R471Q, EDC4 V477M, EDC4 K514R, and EDC4 K1157R cell lines were treated with 0.05 µg/mL diepoxybutane for at least one cell division in culture. Cells were processed by flow cytometry following the procedure previously described[45]. Briefly, cells were stained with ethidium monoazide and subsequently lysed and stained with Sytox green. Samples were kept at 4 °C until flow cytometry acquisition.

**Statistics**. Significance of the functional studies was analyzed using two-tailed Student's $t$ test with the exception of: Fig. 5d, e, Supplementary Fig. 2d, e, and Supplementary Fig. 6d, e, where a one-way analysis of variance was performed followed by a Dunnett's multiple comparison test, and Fig. 2c and Supplementary Fig. 1b, where a Mann–Whitney test was performed due to a lack of adjustment to a normal distribution of the data. A $p$ value of <0.05 was considered statistically significant. RPA32 and RAD51 foci evaluation was done in a blind fashion. A Fisher's exact test was used to compare carrier frequencies of *EDC4* variants between controls and breast cancer cases

**Data availability**. The data that support the findings of this study are available from the corresponding author upon reasonable request.

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

## Acknowledgements

We thank Dr. Andre Nussenzweig and Dr. Elsa Callen for critical reading of the manuscript and helpful comments and Dr. Joaquin Dopazo for his help with the CIBERER Spanish Variant Server. Surrallés laboratory is supported by the ICREA-Academia program, the Marató de TV3 (project 464/C/2012), the Spanish Ministry of Health (projects FANCOSTEM and FANCOLEN), the Spanish Ministry of Economy and Competiveness (projects CB06/07/0023, SAF2012-31881, SAF2013-45836-R and SAF2015-64152-R), the European Commission (EUROFANCOLEN project HEALTH-F5-2012-305421 and P-SPHERE COFUND project), the Fanconi Anemia Research Fund Inc., and the "Fondo Europeo de Desarrollo Regional, una manera de hacer Europa" (FEDER). CIBERER is an initiative of the Instituto de Salud Carlos III, Spain. This work was also supported by grants from the Asociación Española Contra el Cáncer (AECC, Hereditary Cancer group, 2010), the Generalitat de Catalunya (SGR 2014-364 and 2014-338), and the Instituto de Salud Carlos III (PIE13/00022-ONCOPROFILE, CP10/00617, PI10/01422, PI12/02585, PI13/00285, PI15/00854, PI16/00563, PI16/01218, and RTICC RD12/0036/0008). S.G.-E. is funded by a Miguel Servet contract from the Instituto de Salud Carlos III.

## Author contributions

G.H., M.J.R., M.A.P., and J.S. conceived the study. G.H., M.J.R., J.M., M.B., M.A.-C., and R.P.-C. designed and performed experiments and analyzed data. R.P.-C., G.R.d.G., J.F., and N.G. performed genetic analyses. A.L., S.G.-E., O.D., J.B., M.S., A.T., J.B., P.R., P.P., and C.L. contributed to clinical and exome data. X.S.P. analyzed sequencing pool data. D.S. analyzed FA patients. J.S., G.H., and M.J.R. wrote the manuscript and all co-authors reviewed it.

## Additional information

**Competing interests:** The authors declare no competing interests.

