## [Peer Review File · Nature Communications]

Reviewers' comments:

Reviewer #1 (Remarks to the Author):

This paper by Hernandez et al describes a new role for EDC4, a gene encoding a component of the cytoplasmic mRNA decay machinery, in DNA repair and potentially in breast cancer development. Starting with a two-hybrid screen, the authors identify EDC4 as a new interactor of TOPB1 a well known protein involved in genome stability. After confirming that EDC4 interacts with TOPB1 and BRCA1 and BRIP1 (two known interactors of TOPB1) and that EDC4 is present in the nucleus, the authors go on and test whether EDC4 could have a role along with BRAC1 in DNA repair. The authors show that similarly to BRAC1 depleted cells, EDC4 depleted cells display impaired DNA damage response, increased sensitivity to DNA damaging agents and increased genome instability. Using a deletion approach on EDC4, the authors then identify the EDC4 WD-40 protein domain to be critical for the DNA damage repair function and importantly show that this domain does not affect P-body formation suggesting that the role of EDC4 in mRNA decay is independent from its role in DNA repair (although the authors do not definitely show that mRNA decay per se is non-affected). The author then further investigate the role of EDC4 in DNA repair and show that, similarly to BRAC1, EDC4 promotes DNA resection at DSB and is critical to promote homologous recombination. Interestingly, the authors also find that EDC4 depleted cells are sensitive to PARP inhibition. Finally, using a large cohort of breast cancer patients with no mutations in BRAC1, the authors identify potential cancer causing mutations in EDC4 and show that these mutations, found around the critical WD-40 domain of EDC4 cause similar phenotypes as the one observed in EDC4 depleted cells suggesting that these mutations are indeed loss of function mutations susceptible to promote cancer development.

Overall, I find this study interesting. The authors provide a large amount of data describing a new role of EDC4 in DNA repair, genome stability and cancer susceptibility. Although mRNA decay defective cells are known to have increased genome instability, the fact that EDC4 binds to known DNA repair proteins and localizes to DSB is new. The manuscript is concise, well written and easy to follow. The statistical analysis methods used are appropriate, however I have a few comments regarding the data representation in the figures, as some figures legends lacks details on how the statistical analysis was performed, and I would like the author to add a few controls in some of their experiments (see below). Overall, this article can be of broad interest for the DNA damage community, the mRNA decay/P-body community as well as the cancer community.

Major comments

1- Data representation. I suggest replacing the bar graphs with stacked dot-plot with mean \pm SD depicted. This will allow a better representation of the data dispersion and data reproducibility. I also suggest replacing asterisks for statistical support with the exact p-values obtained for each statistical test.

In Fig 2c, Fig S1b, standard deviation should be shown and not SEM. How was the statistical test performed? Was the total number of chromosomal aberration or the number for each category of aberrations compared in the mutant vs control? This should be clarified. It is also unclear how many times this experiment has been repeated. The authors indicate that they analyzed 40 metaphases, but are these metaphases from cells coming from one experiment or more? This should be clarified as well, and the experiment should be repeated at least once if the 40 metaphases came from the same set of cells.

In Fig 2i, no statistical test was shown. The data should be backed up with statistical testing.

In Fig 3c, only one representative experiment is shown. In fig 3d, the results from three independent experiments are combined. Why this inconsistency?

Legend of Fig 3d mentions panel h, there is no panel h in fig. 3.

Fig 3f, is it mean \pm SD? the legend doesn't specify

Fig S2g and S5b. How many replicates were performed for this experiment?

Fig S6c. The data should be backed-up with statistical analysis for significance, how many replicates are included in the data shown?

Fig4d-e and S6d-e, what is the rationale for using an ANOVA corrected by Dunnett's multiple comparison test?

2- Laser microirradiation confocal microscopy experiment: I would like the author to clearly demonstrate that EDC4 relocalization is to known DSB foci using a known DSB binding protein as positive marker in this experiment. Since EDC4 binds to TOPBP1, BRAC1 and BRIP1, I would suggest showing colocalization with at least one of these proteins in this assay.

3- The full result of the two hybrid screen should be provided as supplementary table and should include a list of all the proteins tested.

4- Fig. 2. I fail to understand why the authors included FANCD2 controls but not BRCA1 controls in panels b-d as the focus of the paper is on EDC4 phenocopying BRCA1 loss of function. I suggest the authors include BRCA1 siRNA data in these figures.

5- I would like the authors to demonstrate that the missense mutations that they identify from the patients data do not impair P-body formation. I would also like the authors to test if these mutations render cells sensitive to PARP inhibition as this data could be useful for clinicians.

6- Regarding the following statement:

Finally, several breast cancer susceptibility genes with a role in HR-mediated repair are found to be mutated in FA patients. However, we did not find EDC4 mutations or lack of EDC4 expression in 17 FA patients with normal FANCD2 ubiquitination and without mutations in any of the 22 known FA genes²⁶ (data not shown).

I would suggest this data be provided as a supplementary figure, as Nature Communications requests that authors avoid "data not shown" statements.

7- the section between line 274 and 278 in the Discussion mentions Results that are not presented in the Results section.

8- a supplementary figure including un-cropped images for all the immunoblots should be provided.

Minor comments

1- For improved clarity, I suggest splitting the figures so there is one figure for each Results sub-sections

2- The introduction should include a summary of the results and could present a more detailed view of the role of BRAC1\FA in cancer and chemotherapeutic susceptibility.

3- The subsections in the Methods section should appear in the same order as in the Results section.

4- Antibody dilutions should be indicated in the Methods section.

5- I invite the authors to edit their Methods section carefully as I have noted quite a few wording and sentence construction problems.

6- Fig 4c legend. The legend does not describe the experiment.

7- Fig S2f, the MW of each truncated version of Edc4 should be indicated at the bottom of each line so the reader can actually evaluate where each of these protein variants migrate on the gel.

8- units: digits and units should be separated with a space. I noted a few inconsistencies especially in the Methods section.

9- typos:

line 124: I would replace "aggregate" with "binds"

line 224: identified EDC4 as extremely intolerant gene: "an" is missing

line 244: which is compatible with the hypothesis the Y147C is a rare: "that" is missing

line 487: spelling of inhibitor should be corrected

line 498: inside or close the WD40 domain: "to" is missing

line 651: cytometry is misspelled

line 653: wording is incorrect

line 677: replace 1.300 with 1,300

line 687: culture is misspelled

line 717: from is misspelled

line 764: replace 5.000 and 50.000 with 5,000 and 50,000

Reviewer #2 (Remarks to the Author):

In the manuscript entitled "Decapping protein EDC4 regulates DNA repair and phenocopies BRCA1", Gonzalo Hernandez et al. describe a novel function of EDC4 in DNA damage response, more specifically in Homologous Recombination and resection, independently of its known involvement in mRNA decapping.

EDC4 was found to interact with TOPBP1 in a Y2H screen and here the authors demonstrate its interaction with TOPBP1 and with BRCA1 in human cells.

The authors have searched for EDC4 mutations in a cohort of 300 human breast cancers and found 5 rare missense variants. Subsequent experiments showed that these residues had a crucial impact on EDC4 role in DNA damage response. In addition, in the Discussion section, the authors mention 2 other residues whose post-translational modification is directly implicated in DNA damage response.

General comments:

- This study puts together lots of extremely interesting data and these results are of great interest. This study is certainly another piece in the growing body of evidence that mRNA metabolism is closely linked to DNA repair. However an effort should be made to homogenize the presentation of the data to make it simpler and more precise about the cellular systems. Finally the raw data should be always promoted at the expense of "fold to the control" as controls are also informative.

- Although it is suggested by co-IP experiments in particular and often in the text, there is no clear evidence of whether EDC4 and BRCA1 act in the same pathway -or not- in response to DNA damage. Some double-silencing experiments at least for sensitivities and homologous recombination repair, would be greatly informative to check whether EDC4 and BRCA1 act in the same pathway or if EDC4 just phenocopies BRCA1. That's the only major experimental complement I think is necessary.

- All along the manuscript, DEB or MMC sensitivity experiments are used as ways to measure DNA repair. I don't agree with that way of describing the results and would prefer the term "DNA damage sensitivity/resistance" instead of "DNA repair": cell sensitivity reflects other parameters

than just DNA repair (cell cycle checkpoints, apoptosis, cell growth and metabolism, etc.). In addition it is published in the literature that EDC4 strongly affects cell viability in the absence of DNA damage -which is not surprising but certainly confirms that EDC4 impact of cell survival cannot be attributed solely to its potential implication in DNA repair.

Specific comments:

- "EDC4 physically interacts with TOPBP1 and associates with BRCA1"

Fig 1: all the interaction experiments are performed in non stressed HeLa cells. Did the authors perform the same studies in cells submitted to genotoxic stress? An increased interaction would strengthen the relevance of the formation of this complex.

Fig 1f: Please homogenise BRIP1 and FANCD1. In this figure, the fact that BRCA1 co-IP with EDC4 and FANCD1 does not necessarily mean that FANCD1 and EDC4 are together in a same complex with BRCA1. Several different BRCA1 complexes have been described in the literature, even sometimes with opposite roles. So please modify accordingly the conclusion of this paragraph.

- "EDC4 is involved in DNA damage response"

Figure 2 -and the whole study- combines various cell types: HeLa, U2OS, primary fibroblasts or HEK293T. So please write the name of the cell line on the figures so that the reader knows what it is about. Also please homogenize the way sensitivity assays are represented: "% of survival" (fig 1f) is by far more informative than "survival (ratio vs. siMock(=1))"

Also sometimes survival is shown with histograms (Fig 2b, 2f) and sometimes with curves (2d, 2i). Please homogenize.

Fig 2a: It is not clear in what cells is done this Western Blot and why for each sample, 2 lanes are represented? As numerous cellular systems are used it would be useful to see the depletion in all of them in a supplemental figure for example.

Fig 2d: Result section (line 141) mentions that this figure shows MMC sensitivity but the figure shows DEB sensitivity. Please correct.

Fig 2e: Authors mention in line 117 that EDC4 is present in the nucleus and that's what their cellular fractionation in the absence of damage shows in Fig 1: how come the confocal microscopy with the GFP-fused form of EDC4 is not located in the nucleus before damage? Please comment.

Fig 2g: Please show the real cell cycle distribution. I might have missed it but please explain how the %G2 is calculated.

- "EDC4 regulates HR-mediated repair"

Fig3a and 3b show measurement of HR with the DR-GFP substrate. One technical issue with this substrate is that it strongly relies on the equal expression of I-SceI among samples. Did the authors check that? Also measurements of HR require checking the impact of silencing on cell cycle distribution after I-SceI expression to rule out the hypothesis that HR reduction is due to S/G2 reduction.

Finally the material and Methods section indicate that % of GFP positive cells are normalized to the % of S-phase cells. That is not correct.

After transfection I-SceI is expressed as soon as 5 hours after transfection and cuts already occur at that time. It is still expressed 48h after transfection so induction and repair of damages are both occurring during the 48h time frame.

Should you normalize the reading of GFP positive cells, it has to be with the % of cells expressing I-SceI, which is a mark of transfection efficiency (a critical parameter in these experiments as mentioned above).

In this Figure 3, no indication is given with regards to the dose and duration of chemical treatments or dose and time after irradiation. For immunofluorescence experiments, no non treated controls are shown, and again Fig 3f represents induced RAD51 foci: it is always more informative to show raw data with % of cells with RAD51 foci in non treated vs. treated samples. Here RAD51 could be counted only in G2 cells with a cyclin counterstaining.

The experiments measuring DNA ends resection are introduced in the Results section by the known roles of BRCA1 in stimulating resection. However to be fair, authors should mention that BRCA1 is also part of protection complexes (with RAP80/Abraxas or with Ku80). Finally, did the authors perform sensitivity to PARPi after BRCA1 depletion and after double siRNA for EDC4 and BRCA1?

- "Germline EDC4 mutations in breast cancer patients"

These results are really important. However again the representation in Fig 4d and 4e should be changed for raw data and not "fold to the control".

-Discussion section

To me, the results obtained with the mutants on K514 and K1157 should be moved to the results section. Indeed they again come into support of EDC4 functioning in DNA damage response. Also they should be tested for HR proficiency with the DR-GFP assay. In addition this region corresponds to the preys identified in the Y2H screen: shouldn't they be tested for TOPBP1 interaction?

Only a small remark about the connections between RNA metabolism and DNA repair: some RAD52 and homology mediated repair pathways are mediated through reverse transcription of mRNA. It might be of interest for this study to discuss the possibility that EDC4 would be implicated in that specific type of homology directed repair (you can check for example Mazina et al. Mol Cell 2017). Also interplays between the small RNAs machinery like Ago2, DICER and DROSHA have also been implicated, this time in RAD51 mediated repair : the work of Dr. d'Adda di Fagnaga'group should be mentioned either in the introduction or the discussion.

-typos and small comments :

- line 455: when de WT EDC4

- line 651: analyzed by cell citometry

- in Materials and Methods, please detail DEB as diepoxybutane

- in extended figures 1 and 6: Please blot for appropriate loading controls and not "unspecific bands" or "LC", which we don't know what they are.

- in extended figure 2 : how come the molecular weight of delta 5 is so different from MW of delta 1, 2 or 3, when the size of the deletion is roughly the same?

Reviewer #1

This paper by Hernandez et al describes a new role for EDC4, a gene encoding a component of the cytoplasmic mRNA decay machinery, in DNA repair and potentially in breast cancer development.

Starting with a two-hybrid screen, the authors identify EDC4 as a new interactor of TOPB1 a well known protein involved in genome stability. After confirming that EDC4 interacts with TOPB1 and BRCA1 and BRIP1 (two known interactors of TOPB1) and that EDC4 is present in the nucleus, the authors go-on and test whether EDC4 could have a role along with BRAC1 in DNA repair. The authors show that similarly to BRAC1 depleted cells, EDC4 depleted cells display impaired DNA damage response, increased sensitivity to DNA damaging agents and increased genome instability. Using a deletion approach on EDC4, the authors then identify the EDC4 WD-40 protein domain to be critical for the DNA damage repair function and importantly show that this domain does not affect P-body formation suggesting that the role of EDC4 in mRNA decay is independent from its role in DNA repair (although the authors do not definitely show that mRNA decay per se is non-affected). The author then further investigate the role of EDC4 in DNA repair and show that, similarly to BRAC1, EDC4 promotes DNA resection at DSB and is critical to promote homologous recombination. Interestingly, the authors also find that EDC4 depleted cells are sensitive to PARP inhibition. Finally, using a large cohort of breast cancer patients with no mutations in BRAC1, the authors identify potential cancer causing mutations in EDC4 and show that these mutations, found around the critical WD-40 domain of EDC4 cause similar phenotypes as the one observed in EDC4 depleted cells suggesting that these mutations are indeed loss of function mutations susceptible to promote cancer development.

Overall, I find this study interesting. The authors provide a large amount of data describing a new role of EDC4 in DNA repair, genome stability and cancer susceptibility. Although mRNA decay defective cells are known to have increased genome instability, the fact that EDC4 binds to known DNA repair proteins and localizes to DSB is new. The manuscript is concise, well written and easy to follow. The statistical analysis methods used are appropriate, however I have a few comments regarding the data representation in the figures, as some figures legends lacks details on how the statistical analysis was performed, and I would like the author to add a few controls in some of their experiments (see below). Overall, this

article can be of broad interest for the DNA damage community, the mRNA decay/P-body community as well as the cancer community.

We are glad for these very positive comments on our work.

Major comments

1- Data representation. I suggest replacing the bar graphs with stacked dot-plot with mean +/- SD depicted. This will allow a better representation of the data dispersion and data reproducibility. I also suggest replacing asterisks for statistical support with the exact p-values obtained for each statistical test.

To address this reviewer's suggestion, we have replaced the bar graphs with stacked dot-plot in all possible cases (see new Figures 3d, 4b, 4d, 4e, 5d, and 5e and supplemental figures 1a, 1d, 2d, 2e, 6d and 6e). Similarly, we replaced asterisks by the exact p-values as suggested by the reviewer.

In the case of Figure 2d and 3f (fig 4f in the revised manuscript), we added the exact p-values in the corresponding legend with the following texts:

“Statistical analysis comparing the means of the WT cell line with the samples where performed with the following p-values: FANCC^{-/-} (p=0.001;p=0.001), EDC4^{-/-} (p=0.000;p=0.000) and EDC4 corrected (p=0.613;p=0.173).” And “Significance of the differences of the means was analyzed using 2-tailed Student's t test comparing to the Mock: siBRCA2(p=0.0124; p=0.0023; p<0.0001; p=0.0815), siEDC4(p=0.8183; p=0.3206; p<0.0001; p=0.2856).”

In Fig 2c, Fig S1b, standard deviation should be shown and not SEM. How was the statistical test performed? Was the total number of chromosomal aberration or the number for each category of aberrations compared in the mutant vs control? This should be clarified. It is also unclear how many times this experiment has been repeated. The authors indicate that they analyzed 40 metaphases, but are these metaphases from cells coming from one experiment or more? This should be clarified as well, and the experiment should be repeated at least once if the 40 metaphases came from the same set of cells.

We completely agree with the referee that SD is better than SEM. In fact, in most of the graphs of this paper we represent SD. However, in these two graphs we have chosen SEM due to the nature of the biomarker used. Breaks/cell is a biomarker where in most of the cells the value is “0”. But when a DNA repair gene is inhibited, only a fraction of the cells have one or more

breaks leading to large SD masking the clear and statistically significant increase of breaks/cell.

To address the reviewer's comments and clarify Fig2c and FigS1b, we have added the following text in the corresponding legends:

Fig2c

"Data shown represent results from the analysis of 40 metaphases (two experiments of 20 metaphases each one). Error bars indicate mean \pm s.e.m. The statistical test performed in each inhibition to compare breaks/cell (0 vs 0.1 $\mu\text{g}/\text{mL}$ DEB) was a Mann-Whitney test".

FigS1b

Data shown represent results from the analysis of 40 metaphases (two experiments of 20 metaphases each one). Error bars indicate mean \pm s.e.m. The statistical test performed in each inhibition to compare chromatid exchange/cell (0 vs 0.1 $\mu\text{g}/\text{mL}$ DEB) was a Mann-Whitney test.

In Fig 2i, no statistical test was shown. The data should be backed up with statistical testing.

We agree with this comment and apologize for the lacking information. To address this point we have added the following paragraph in the legend of original figure 2i (figure 3f in the revised manuscript):

"Means were statistically compared using the 2-tailed Student's t test. No statistically significant difference was observed between WT, $\Delta 4$ ($p=0.233$; $p=0.516$), $\Delta 6$ ($p=0.852$; $p=0.455$) and corrected cells ($p=0.8832$; $p=0.4646$) while the rest of the mutants were found statistically different compared with the WT cell line (EDC4^{-/-}, $p=0.0013$ and $p=0.0027$; $\Delta 1$, $p=0.001$ and $p=0.002$; $\Delta 2$, $p=0.0009$ and $p=0.0017$; $\Delta 3$, $p=0.0542$ and $p=0.0032$; $\Delta 5$, $p=0.0001$ and $p<0.0001$)."

In Fig 3c, only one representative experiment is shown. In fig 3d, the results from three independent experiments are combined. Why this inconsistency?

We completely agree that it is not required to show two graphical representations of the same data. Accordingly, we removed the bar graph and, instead of showing a scatterplot for only one of the experiments, we now show a completely renewed figure (4c in the revised version) with the combined results of the three experiments. In addition, we added the following text to the

corresponding legend: “Statistical analysis was performed using a Mann-Whitney test”.

Legend of Fig 3d mentions panel h, there is no panel h in fig. 3.

We apologize for this error. This sentence has been removed from the legend.

Fig 3f, is it mean +/- SD? the legend doesn't specify

Yes, it is. This information is now included in the corresponding legend with the following text: “Error bars indicate mean \pm s.d.”

Fig S2g and S5b. How many replicates were performed for this experiment?

This experiment was repeated twice with similar results but we showed only one representative experiment. To avoid misunderstandings, we removed these two graphs and add the ratio of EDC4 expression in the same figure showing the Western blot. In addition, the following sentence was added in the corresponding figure: “This experiment was performed twice and we show a representative Western blot”.

Fig S6c. The data should be backed-up with statistical analysis for significance, how many replicates are included in the data shown?

To address this comment, we have added the following paragraph in the legend of figure S6C:

“Means were statistically compared using the 2-tailed Student's t test. No statistical difference was observed between WT, S741A(p=0.214; p=0.393) and corrected cells (p=0.751; p=0.824) while the ubiquitination mutants were found statistically different compared with the WT cell line (K514R, p=0.001 and p=0.000; K1157R, p=0.000 and p=0.000).”

Fig4d-e and S6d-e, what is the rationale for using an ANOVA corrected by Dunnett's multiple comparison test?

We chose this statistic test because we aimed to compare each mutated/deficient EDC4 cell line with the corrected counterpart and Dunnett's correction allows comparing several cell lines with a unique control in a single analysis avoiding uninformative comparisons.

2- Laser microirradiation confocal microscopy experiment: I would like the author to clearly demonstrate that EDC4 relocalization is to known DSB foci using a known DSB binding protein as positive marker in this experiment. Since EDC4 binds to TOPBP1, BRAC1 and BRIP1, I would suggest showing colocalization with at least one of these proteins in this assay.

We completely agree with this point. To address this reviewer's suggestion, we bought a plasmid containing a cherry-tagged BRCA1 and tried several times to co-express GFP-EDC4 and Cherry-BRCA1 to see co-localization. However, these experiments failed probably due to cell toxicity. In fact, we have tested BRCA1-cherry and PCNA-cherry as a DNA damage marker but both proteins did not behaved as expected. Given the limited time to respond and the strong evidences that we show on the EDC4 and TOPBP1-BRCA1 interaction (yeast two hybrid with 4 independent preys and endogenous and exogenous co-IP experiments shown in Fig 1), we believe that our conclusions are well supported by the currently available data.

3- The full result of the two hybrid screen should be provided as supplementary table and should include a list of all the proteins tested.

The only additional TOPBP1-interacting protein that we unequivocally identified in the yeast-two-hybrid screen was PPHLN1 but it was not further studied in detail and, therefore, we did not mention it in the manuscript. We leave the final inclusion of this apparently irrelevant protein to the editor's discretion.

4- Fig. 2. I fail to understand why the authors included FANCD2 controls but not BRCA1 controls in panels b-d as the focus of the paper is on EDC4 phenocopying BRCA1 loss of function. I suggest the authors include BRCA1 siRNA data in these figures.

The experiment suggested by this reviewer is actually shown in Fig 3. We initially included FANCD2 as a Fanconi anemia/BRCA pathway control protein because the siRNA experiments using FANCD2 were very well standardized in our laboratory since 2007 (see Bogliolo et al., EMBO J, 2007) and we wanted to gain confidence on the role of EDC4 in DNA damage response. In addition, in terms of cellular sensitivity to interstrand cross-linking drugs such as MMC, BRCA1 and FANCD2 function in the same epistatic pathway and both BRCA1 (FANCS) and FANCD2 are mutated in Fanconi anemia patients.

5- I would like the authors to demonstrate that the missense mutations that they identify from the patients data do not impair P-body formation.

To address this reviewer's comment, we have generated GFP fusions of all missense EDC4 mutants (including those identified in breast cancer patients and the ubiquitination mutants) and studied the formation of P-bodies by confocal microscopy. We observed that none of the mutants affected P-body formation. The following sentence is included in the revised version: "We then generated GFP-fusions of the patient-derived missense mutations and observed that they do not impair P-body formation (Supplementary Fig. 5b)" in the Results section, subsection "Germline EDC4 mutations in breast cancer patients". In addition the following sentence has been added in the discussion section, second paragraph: "We then generated GFP-fusions of these ubiquitination mutants and observed that they do not impair P-body formation (Supplementary Fig. 5b)".

We added the following text in the legend for the corresponding Supplementary Fig. 5b:

"Confocal images of cells expressing GFP-fused forms of EDC4 WT, missense mutations found in BRCA1 patients and point mutations affecting lysines required for ICL-repair."

6- Regarding the following statement: "Finally, several breast cancer susceptibility genes with a role in HR-mediated repair are found to be mutated in FA patients. However, we did not find EDC4 mutations or lack of EDC4 expression in 17 FA patients with normal FANCD2 ubiquitination and without mutations in any of the 22 known FA genes²⁶ (data not shown)". I would suggest this data be provided as a supplementary figure, as Nature Communications requests that authors avoid "data not shown" statements.

A table containing the results of the sequencing of the patients has been included in the paper as Supplementary Table 2. Given that no pathogenic mutations were identified in these Fanconi anemia patients, we only list the observed non-pathogenic variants (polymorphisms) in this table. The WB showing that all the studied patients have normal EDC4 expression are shown in Supplementary Figure 7. When collecting these data we realized that, collectively, EDC4 was studied in 17 unassigned FA patients but only 15 of them by DNA sequencing and 12 of them by Western blot, depending on availability of DNA and cell lines, respectively. We apologize for this error. The following text amending this error is now included in the revised version (results section, last paragraph)

“Finally, several breast cancer susceptibility genes with a role in HR-mediated repair are found to be mutated in FA patients. To investigate whether EDC4 is also mutated in FA, we studied DNA or cell lines from 17 FA patients with normal FANCD2 ubiquitination and without mutations in any of the 22 known FA genes²⁶. We did not find EDC4 mutations in 15 patients’ DNAs or lack of EDC4 expression in 12 FA cell lines (Supplementary Table 2 and Supplementary Fig. 7)”.

We added the following lines in the corresponding legends for Supplementary Fig. 7:

“Immunoblots showing that EDC4 is expressed in the studied FA patients. A EDC4/Tubulin ratio is included as a loading control.”

7- the section between line 274 and 278 in the Discussion mentions Results that are not presented in the Results section.

To address this point, we have added the results of the phosphorylation and ubiquitination mutants to the Results section of the paper and kept the discussion of these results in the Discussion section.

The following text is now included in the revised version as the last paragraph of the section “*EDC4 is involved in DNA damage response*”:

“Moreover, it is reported that EDC4 is post-translational modified by lysines 514 and 1157 ubiquitination and by serine 741 phosphorylation in response to DNA damage¹⁰ (Supplementary Fig. 6a). We produced mutations of these residues (Supplementary Fig. 6b) and we observed that cell lines with mutations K514R and K1157R in EDC4 are sensitive to DEB agent and this treatment induced chromosome fragility (increase in MN frequency) and G2 arrest (Supplementary Fig. 6c, d and e) while the S741A mutant does not show DEB sensitivity (Supplementary Fig. 6c). These results indicate that the ubiquitination of EDC4 in lysine K514 and K1157 is crucial for DNA damage resistance.”

8- a supplementary figure including un-cropped images for all the immunoblots should be provided.

We have gathered the uncropped images for the immunoblots into a file that is attached to this re-submission (in PDF format). We leave the final inclusion of this file as Supplementary data/file to the editor’s discretion.

Minor comments

1- For improved clarity, I suggest splitting the figures so there is one figure for each Results sub-sections

We have split Figure 2 in two. Panels a to d are now included in Figure 2. Panels e to j are now included in the new Figure 3. The rest of the Figures and text has been relabeled according to the new panel order.

2- The introduction should include a summary of the results and could present a more detailed view of the role of BRCA1/FA in cancer and chemotherapeutic susceptibility.

To address this point, we have added the following text at the end of the introduction paragraph:

“Here we observe that EDC4, besides its known role in P-bodies, interacts with BRCA1 and is involved in HR-mediated DNA repair by regulating its end-resection step and that germline mutations in EDC4 may confer increased risk of breast cancer. Taken together our results suggest that EDC4 is a functional phenocopy of BRCA1 that could be targeted in cancer therapeutics.”

3- The subsections in the Methods section should appear in the same order as in the Results section.

We have modified the Methods section according to this comment.

4- Antibody dilutions should be indicated in the Methods section.

We have added this information in “Antibodies” (Methods section).

5- I invite the authors to edit their Methods section carefully as I have noted quite a few wording and sentence construction problems.

We have carefully revised the Methods section and all changes are indicated in red color in the revised version of the manuscript

6- Fig 4c legend. The legend does not describe the experiment.

In order to describe the experiment, we have added the following paragraph in Fig 4c legend (Fig. 5c in the revised manuscript):

“Graph shows survival after DEB treatment of five EDC4 mutants generated in HEK293T cells. Data shown represent results from at least two combined independent experiments. Error bars indicate mean \pm s.d. Means were statistically compared using the 2-tailed Student’s t test: EDC4-/- (p=0.000; p=0.000), EDC4 corrected (p=0.760; p=0.748), G42E (p=0.002; p=0.005), S277R (p=0.024; p=0.011), D361E (p=0.001; p=0.001), R471Q (p=0.010; p=0.017) and V477M (p=0.000; p=0.000).”

7- Fig S2f, the MW of each truncated version of Edc4 should be indicated at the bottom of each line so the reader can actually evaluate where each of these protein variants migrate on the gel.

We completely agree with the reviewer. We have modified the panel and moved the size of the mutants to the bottom of the blot. As mentioned above, we have also removed panel FigS2g and added a EDC/ub ratio as a loading control in the panel FigS2f. The new legend for FigS2f reads as follows:

“Immunoblots showing the levels of expression of HA-tagged EDC4 deletion mutants in HEK293T cells with an anti-HA antibody. Molecular weight is indicated at the bottom of the panel; u.b.: unspecific band that serves as internal loading control. Ratio between EDC4 and the unspecific band serves as a loading control. This experiment was performed twice and we show a representative Western blot.”

8- units: digits and units should be separated with a space. I noted a few inconsistencies especially in the Methods section.

We apologize by these errors. We have carefully checked the manuscripts and corrected them.

9- typos:

line 124: I would replace “aggregate” with “binds”

line 224: identified EDC4 as extremely intolerant gene: “an” is missing

line 244: which is compatible with the hypothesis the Y147C is a rare: “that” is missing

line 487: spelling of inhibitor should be corrected

line 498: inside or close the WD40 domain: “to” is missing

line 651: cytometry is misspelled

line 653: wording is incorrect

line 677: replace 1.300 with 1,300

line 687: culture is misspelled

line 717: from is misspelled

line 764: replace 5.000 and 50.000 with 5,000 and 50,000

We thank the reviewer for detecting these typographical errors. All of them have been corrected as suggested

Reviewer #2

In the manuscript entitled "Decapping protein EDC4 regulates DNA repair and phenocopies BRCA1", Gonzalo Hernandez et al. describe a novel function of EDC4 in DNA damage response, more specifically in Homologous Recombination and resection, independently of its known involvement in mRNA decapping.

EDC4 was found to interact with TOPBP1 in a Y2H screen and here the authors demonstrate its interaction with TOPBP1 and with BRCA1 in human cells.

The authors have searched for EDC4 mutations in a cohort of 300 human breast cancers and found 5 rare missense variants. Subsequent experiments showed that these residues had a crucial impact on EDC4 role in DNA damage response. In addition, in the Discussion section, the authors mention 2 other residues whose post-translational modification is directly implicated in DNA damage response.

General comments:

- This study puts together lots of extremely interesting data and these results are of great interest. This study is certainly another piece in the growing body of evidence that mRNA metabolism is closely linked to DNA repair. However an effort should be made to homogenize the presentation of the data to make it simpler and more precise about the cellular systems. Finally the raw data should be always promoted at the expense of "fold to the control" as controls are also informative.

We are grateful to this reviewer for the positive words on our manuscript. To address the reviewer's suggestion, we have homogenized the figures (see our response to reviewer 1) and changed the data representation to use raw data whenever possible (with only one exception as explained later).

- Although it is suggested by co-IP experiments in particular and often in the text, there is no clear evidence of whether EDC4 and BRCA1 act in the same pathway -or not- in response to DNA damage. Some double-silencing experiments at least for sensitivities and homologous recombination repair, would be greatly informative to check whether EDC4 and BRCA1 act in the same pathway or if EDC4 just phenocopies BRCA1. That's the only major experimental complement I think is necessary.

We completely agree that this experiment would provide functionally important information. Accordingly we have performed the suggested survival and HR studies and the results are included in Fig. 4g, Fig. 4h and Supplementary Fig.3d and 3e of the revised manuscript.

Considering the results obtained, we changed the name of the results section from "*EDC4 regulates HR-mediated repair*" to "*EDC4 regulates HR-mediated repair and works with BRCA1 in the same pathway*".

We have added the following text to the results before the last paragraph of the section "*EDC4 regulates HR-mediated repair*":

"Considering the fact that EDC4 and BRCA1 share common phenotypes, we studied the possibility that EDC4 and BRCA1 work in the same pathway. We analyzed the effect of the simultaneous inhibition of EDC4 and BRCA1 in terms of DEB sensitivity and HR repair efficiency. Depletion of EDC4 and BRCA1 renders HeLa cells sensitive to DEB treatment in a similar way to the individual inhibition of EDC4 or BRCA1 (Fig. 4g and Supplementary Fig. 3d). We previously established that EDC4 shows a mild impairment in HR efficiency compared to BRCA1 (Fig. 4b). In this scenario, the inhibition of BRCA1 in cells inhibited for EDC4 shows impairment in repair similar to the sole inhibition of BRCA1 (Fig. 4h and Supplementary Fig. 3e)."

We also added the following text to the last paragraph of the same section: "the same pathway of".

The following text was added to the legends in Figure 4 and Supplementary Figure 3.

Fig. 4g: "Simultaneous BRCA1 and EDC4 depletion in HeLa cells have a similar functional impact that individual inhibition of EDC4 or BRCA1 after DEB treatment. Data shown represent results from three combined independent experiments. Error bars indicate mean \pm s.d. Significance of the differences of

the means was analyzed using 2-tailed Student's t test comparing to the Mock: siEDC4(p=0.0181;p=0.0231), siBRCA1(p=0.0158;p=0.0188) and siBRCA1-siEDC4 (p=0.0423;p=0.0341)."

Fig. 4h: "Quantification of the remaining HR repair of BRCA1, EDC4 and BRCA1/EDC4 siRNA-silenced U2OS-DR-GFP cells. Data shown represent results from at least two combined independent experiments. Error bars indicate mean \pm s.d."

Supplementary Fig.3d: "Immunoblots showing the inhibition efficiency of EDC4, BRCA1 and the simultaneous inhibition of EDC4 and BRCA1 in HeLa cells."

Supplementary Fig.3e: "Immunoblots showing the inhibition efficiency of EDC4, BRCA1 and the simultaneous inhibition of EDC4 and BRCA1 in U2OS-DR-GFP cells by siRNA."

- All along the manuscript, DEB or MMC sensitivity experiments are used as ways to measure DNA repair. I don't agree with that way of describing the results and would prefer the term "DNA damage sensitivity/resistance" instead of "DNA repair": cell sensitivity reflects other parameters than just DNA repair (cell cycle checkpoints, apoptosis, cell growth and metabolism, etc.). In addition it is published in the literature that EDC4 strongly affects cell viability in the absence of DNA damage -which is not surprising but certainly confirms that EDC4 impact of cell survival cannot be attributed solely to its potential implication in DNA repair.

To address this reviewer's comment, we have changed DNA repair to DNA damage response/sensitivity in the text as indicated in the revised manuscript in red color. However, once the implication of EDC4 in HR repair has been proven, we consider that the use of the term "DNA repair" is appropriated.

Specific comments:

-"EDC4 physically interacts with TOPBP1 and associates with BRCA1"

Fig 1: all the interaction experiments are performed in non stressed HeLa cells. Did the authors perform the same studies in cells submitted to genotoxic stress? An increased interaction would strengthen the relevance of the formation of this complex.

We indeed performed interaction experiments after genotoxic stress (MMC and HU) and observed a small increase in the interaction when compared to untreated cultures. However, the differences are small and we decided not to include them in order to focus the paper only in the most conclusive results and,

given the limited number of words and figure, to save space. For the reviewer's information, we include the following blot showing a small increase in the interactions between EDC4 and BRCA1 or TOPBP1 upon HU treatment (l.e and h.e mean "low exposure" and "high exposure", respectively):

Fig 1f: Please homogenise BRIP1 and FANCI.

We have changed FANCI for BRIP1 in the figure and corresponding legend in line 803.

In this figure, the fact that BRCA1 co-IP with EDC4 and FANCI does not necessarily mean that FANCI and EDC4 are together in a same complex with BRCA1. Several different BRCA1 complexes have been described in the literature, even sometimes with opposite roles. So please modify accordingly the conclusion of this paragraph.

-"EDC4 is involved in DNA damage response"

Figure 2 -and the whole study- combines various cell types: HeLa, U2OS, primary fibroblasts or HEK293T. So please write the name of the cell line on the figures so that the reader knows what it is about.

We have introduced the names of the cell types in the corresponding legends in lines: 808, 840, 867, 989, 905, 906, 921 and 928.

Also please homogenize the way sensitivity assays are represented: "% of survival" (fig 1f) is by far more informative than "survival (ratio vs. siMock(=1))"

To address this point, we have changed the representation of the data to "% of survival" as suggested by the reviewer

Also sometimes survival is shown with histograms (Fig 2b, 2f) and sometimes with curves (2d, 2i). Please homogenize.

We have changed from histograms to curves for all the survivals. Due to lack of space in the graph, and added the statistical analysis of the data in the corresponding legends with the following texts.

Figure 2b:

"Statistical analysis comparing the means of the Mock-treated versus siRNA-treated samples where performed with the following p-values: siFANCD2 (10 nM, p=0.0376; 25 nM, p=0.0114), siEDC4-1 (10 nM, p=0.0747; 25 nM, p=0.0018), siEDC4-2 (10 nM, p=0.0227; 25 nM, p=0.0199), siEDC4-3 (10 nM, p=0.2004; 25 nM, p=0.0028)."

Figure 3c in the revised manuscript (Fig 2f in the original):

"Statistical analysis comparing the means of the Mock-treated versus gene-specific siRNA-treated samples for the sensitivity to DEB where performed with the following p-values: siBRCA1 (10 nM, p=0.0122; 25 nM, p=0.0005), siEDC4 (10 nM, p=0.0115; 25 nM, p=0.0012), siDCP1a (10 nM, p=0.1574; 25 nM, p=0.1862)."

Fig 2a: It is not clear in what cells is done this Western Blot and why for each sample, 2 lanes are represented? As numerous cellular systems are used it would be useful to see the depletion in all of them in a supplemental figure for example.

This Western Blot was performed in HeLa cells. We have introduced this information in the corresponding figure legend of the revised manuscript. We have explained that the two lanes represented per sample indicate two replicates per inhibition. We have introduced in a renewed Supplementary figure 3 all the Western blots showing the siRNA mediated depletion in all the cellular systems that we have used.

Fig 2d: Result section (line 141) mentions that this figure shows MMC sensitivity but the figure shows DEB sensitivity. Please correct.

We apologize for this error that has been changed accordingly.

Fig 2e: Authors mention in line 117 that EDC4 is present in the nucleus and that's what their cellular fractionation in the absence of damage shows in Fig 1: how come the confocal microscopy with the GFP-fused form of EDC4 is not located in the nucleus before damage? Please comment.

In the analysis of confocal microscopy we have observed cells with EDC4-GFP located in the nucleus but the percentage is variable between experiments. To address this reviewer's comment, an example of *in vivo* nuclear localization of EDC4-GFP in untreated cells has been added to Figure 3a (new panel).

The following text has been added in the Fig.3 legend:

"Confocal image of U2OS cells expressing GFP-tagged wild-type EDC4 showing an example of cell with nuclear localization of EDC4 in the absence of DNA damage."

Fig 2g: Please show the real cell cycle distribution. I might have missed it but please explain how the %G2 is calculated.

To address this point, we included a new panel (Supplementary Figure 8) with the actual cell cycle distributions of treated and untreated cells upon siRNA-based inhibition of EDC4, BRCA1 and DCP1a. We also added the following line in the corresponding legend:

"Histograms showing the cell cycle distribution of Mock, BRCA1, EDC4 and DCP1a HeLa depleted cells and the effect of MMC treatment on the amount of cells arrested in the G2 phase of the cell cycle."

To clarify how the %G2 cells is calculated, we added the following paragraph in Material and Methods section of the revised Manuscript "The percentage of G2 cells was calculated from the cytometer data using the FlowJo program. When the intensity of the propidium Iodide was evaluated in a histogram, the G2 pick was selected and the percentage of cells in G2 was calculated related to the total amount of cells that were evaluated for cell cycle distribution."

-"EDC4 regulates HR-mediated repair"

Fig3a and 3b show measurement of HR with the DR-GFP substrate. One technical issue with this substrate is that it strongly relies on the equal expression of I-SceI among samples. Did the authors check that? Also measurements of HR require checking the impact of silencing on cell cycle distribution after I-SceI expression to rule out the hypothesis that HR reduction is due to S/G2 reduction.

Finally the material and Methods section indicate that % of GFP positive cells are normalized to the % of S-phase cells. That is not correct.

After transfection I-SceI is expressed as soon as 5 hours after transfection and cuts already occur at that time. It is still expressed 48h after transfection so induction and repair of damages are both occurring during the 48h time frame.

Should you normalize the reading of GFP positive cells, it has to be with the % of cells expressing I-SceI, which is a mark of transfection efficiency (a critical parameter in these experiments as mentioned above).

We completely agree with the referee that controlling the cell cycle distribution and the equal expression of I-SceI among samples is essential in the HR/DR-GFP assay and these experimental variables were indeed taken in to account: as already indicated in the original version of the manuscript, cells were transfected with an I-SceI endonuclease expression vector (pCBASce) or with an empty vector (pCAGGS) or with a plasmid constitutively expressing GFP (pNZE-GFP) after siRNA inhibition. The percentage of GFP positive cells obtained with this last plasmid was used to normalize the efficiency of transfection in cells transfected with the I-SceI endonuclease expressing plasmid. Transfection with the empty vector was used to substrate background fluorescence. The cell cycle distribution was determined by flow cytometry after staining cells with propidium Iodide and the percentage of GFP positive cells was corrected related to cell cycle distribution after each siRNA inhibition.

In this Figure 3, no indication is given with regards to the dose and duration of chemical treatments or dose and time after irradiation.

We have added this information in the corresponding figure legend now labeled Figure 4 in lines 866, 869 and 872.

For immunofluorescence experiments, no non treated controls are shown, and again Fig 3f represents induced RAD51 foci: it is always more informative to show raw data with % of cells with RAD51 foci in non treated vs. treated samples.

We completely agree with the referee that with the use of the raw data the results are usually more informative. We changed this panel to show the % of cells in both untreated and treated cells and compared the inductions between the different siRNA.

Here RAD51 could be counted only in G2 cells with a cyclin counterstaining.

We used the Clik-iT EdU from Invitrogen as standard method. EdU is an analog of thymidine and is incorporated into DNA during DNA synthesis. We analyzed RAD51 foci only in EdU positive cells to avoid variations in foci formation due to variations in cell cycle distribution. This was already explained in the original version of the manuscript.

The experiments measuring DNA ends resection are introduced in the Results section by the known roles of BRCA1 in stimulating resection. However to be fair, authors should mention that BRCA1 is also part of protection complexes (with RAP80/Abraxas or with Ku80).

We agree with the reviewer. We have added the following text before the last paragraph of the Discussion section:

“Given that EDC4 interacts with BRCA1, it is probably a member of at least one of the multiple BRCA1-containing protein complexes: BRCA1-A, BRCA1-B, BRCA1-C, BRCA1/PALB2/BRCA2 and the BRCA1/BARD1 heterodimer³⁴. Each of these complexes has different functions related to DNA repair and cell cycle checkpoint activation and some of these complexes have antagonistic functions. BRCA1-B and BRCA1-C are known to promote DSB end resection¹⁸, thus promoting HR, but BRCA1-A limits it resulting in a reduction in HR repair³⁵.”

Finally, did the authors perform sensitivity to PARPi after BRCA1 depletion and after double siRNA for EDC4 and BRCA1?

We did not perform the sensitivity to PARPi in BRCA1 depleted cells but instead we targeted BRCA2 by siRNA in the PARPi sensitivity. Given that both BRCA1 and BRCA2 participate in the same pathway, we would expect similar results in BRCA1 depleted cells.

- "Germline EDC4 mutations in breast cancer patients"

These results are really important. However again the representation in Fig 4d and 4e should be changed for raw data and not "fold to the control".

The baseline frequencies of MN and G2 arrest in the seven cells lines (EDC4 KO, corrected and the five mutations) are highly variable as they derive from single clones that were selected to ensure an expression of mutant EDC4 similar to the expression of the endogenous protein. To measure genetic complementation of DDR deficiency, what it is important are not the absolute number but the mutagen-induced increase in DDR markers (MN and G2 arrest). For this reason, in this specific case we prefer to keep the original format of the Y axes in these two figures (5d and 5e in the revised manuscript)

-Discussion section

To me, the results obtained with the mutants on K514 and K1157 should be moved to the results section. Indeed they again come into support of EDC4 functioning in DNA damage response. Also they should be tested for HR proficiency with the DR-GFP assay. In addition this region corresponds to the preys identified in the Y2H screen: shouldn't they be tested for TOPBP1 interaction?

Mutants K514 and K1157 are mutated in points where the endogenous protein suffers post-translational modifications. We have observed that these ubiquitination sites are important for the function of EDC4 in DNA damage response. For this reason, and in agreement with the reviewer's comment, we have moved these results to the results section of the revised manuscript. We agree that there are a number of open questions yet to be investigated regarding our initial discovery but we consider out of the scope of the present work: are these posttranslational modifications essential for the EDC4 interactions? Which is the ubiquitin ligase responsible? How are these posttranslational modifications regulated? Are they druggable targets??. These questions are under investigation in our laboratory and we will require at least two additional years of extensive research to get definitive answers.

Only a small remark about the connections between RNA metabolism and DNA repair: some RAD52 and homology mediated repair pathways are mediated through reverse transcription of mRNA. It might be of interest for this study to discuss the possibility that EDC4 would be implicated in that specific type of homology directed repair (you can check for example Mazina et al. Mol Cell 2017).

We thank the reviewer for this interesting suggestion. To address this point, we have added a sentence in this direction in the Discussion section (line 300-302, paragraph 1) as follows: "Given the dual role of EDC4, it would be interesting to investigate whether EDC4 plays a role in homology directed repair using RNA as a template for DSB repair in a way similar to Rad52³³."

Also interplays between the small RNAs machinery like Ago2, DICER and DROSHA have also been implicated, this time in RAD51 mediated repair: the work of Dr. d'Adda di Fagagna'group should be mentioned either in the introduction or the discussion.

We have added the following brief comment on this topic in the Discussion section paragraph 1: "It is also known that several proteins involved in the smallRNA machinery like DROSHA and DICER affect the recruitment of certain DNA damage response factors like MDC1 and 53BP132, thus reinforcing the idea of an interplay between DNA repair and RNA metabolism.". In relation to this point we have cited the following paper: "Francia S, Cabrini M, Matti V, Oldani A, d'Adda di Fagagna F.J. DICER, DROSHA and DNA damage response RNAs are necessary for the secondary recruitment of DNA damage response factors. Cell Sci. 2016 Apr 1;129(7):1468-76. doi: 10.1242/jcs.182188. Epub 2016 Feb 16".

-typos and small comments:

- line 455: when de WT EDC4.

Corrected

- line 651: analyzed by cell citometry

Corrected

- in Materials and Methods, please detail DEB as diepoxybutane

Corrected

- in extended figures 1 and 6: Please blot for appropriate loading controls and not "unspecific bands" or "LC", which we don't know what they are.

We agree that is better to use the appropriate loading controls (vinculin) but we only resorted to use unspecific bands when forced by technical problems with the blotting of vinculin. Given that the results are clear and the amount of protein irrelevant, we opted to keep the format of this blot as it was in the original version of the manuscript.

- in extended figure 2: how come the molecular weight of delta 5 is so different from MW of delta 1, 2 or 3, when the size of the deletion is roughly the same?

We do not have an experimentally proved explanation but EDC4 is predicted to be a highly modified protein in terms of phosphorylations and, to a lesser extent, ubiquitinations. The deletion of large regions of EDC4 could impair these modifications affecting the electrophoretic mobility of the protein.

I hope that we satisfactorily addressed all the comments raised by the reviewers and that our paper is now finally acceptable for publication in Nature Communications.

I am looking forward to hearing from you at your earliest convenience.

Sincerely,

Dr. Jordi Surrallés
Director of the Genetics Department, Hospital de Sant Pau
Full Professor of Genetics, Universitat Autònoma de Barcelona

REVIEWERS' COMMENTS:

Reviewer #1 (Remarks to the Author):

The authors have carefully addressed all of my concerns. It is unfortunate that the co-localization analysis did not prove fruitful, but I agree that the protein-protein interaction data presented provide reasonable support for their conclusions, as does the newly added double knock-down data.

Reviewer #2 (Remarks to the Author):

Thanks to Hernandez and coll. for their response to my comments.

There is only one small typo about figure 4g where it is stated in the text that it is DEB survival when the figures indicates MMC.

Other than that, the manuscript is in my opinion now fully suitable for publication in Nature Communications.